# Infinity-Parser: Layout-Aware Reinforcement Learning for Scanned Document Parsing

## Abstract

Document parsing from scanned images into structured formats remains a significant challenge due to its complexly intertwined elements such as text paragraphs, figures, formulas, and tables. Existing supervised fine-tuning methods often struggle to generalize across diverse document types, leading to poor performance, particularly on out-of-distribution data. This issue is further exacerbated by the limited availability of high-quality training data for layout-aware parsing tasks. To address these challenges, we introduce LayoutRL, a reinforcement learning framework that optimizes layout understanding through composite rewards integrating normalized edit distance, paragraph count accuracy, and reading order preservation. To support this training, we construct the Infinity-Doc-400K dataset, which we use to train Infinity-Parser, a vision-language model demonstrating robust generalization across various domains. Extensive evaluations on benchmarks including OmniDocBench, olmOCR-Bench, PubTabNet, and FinTabNet show that Infinity-Parser consistently achieves state-of-the-art performance across a broad range of document types, languages, and structural complexities, substantially outperforming both specialized document parsing systems and general-purpose vision-language models. We will release our code, dataset, and model to facilitate reproducible research in document parsing.

## 1 Introduction

Document parsing aims to convert scanned documents into structured, machine-readable formats and represents one of the core tasks in document intelligence (Hwang et al., 2021; Wang et al., 2024b; Wei et al., 2024; Xia et al., 2024; Zhang et al., 2024). Unlike traditional OCR that focuses solely on text recognition, document parsing requires comprehensive recovery of hierarchical document structures, including the dependency relationships among elements such as paragraphs, headers, tables, and formulas—a capability that is crucial for downstream applications including legal contract analysis, scientific literature mining, and financial report processing. Traditional approaches typically rely on multi-stage pipelines that decompose the task into supervised sub-tasks—such as layout detection, OCR, table recognition, and formula recognition—followed by heuristic post-processing to reconstruct document structure (Blecher et al., 2024; Wei et al., 2025; Liu et al., 2024b; Wei et al., 2024; Bai et al., 2024; Chen et al., 2024). However, such pipeline-based methods are prone to error propagation and exhibit limited adaptability when confronted with diverse layout variations.

Recent approaches primarily reformulate document parsing as end-to-end perception tasks using vision-language models (VLMs) trained through supervised fine-tuning (SFT). However, this paradigm faces fundamental limitations. Although SFT provides token-level supervision, it often overfits to surface patterns rather than learning generalizable structural representations. This limitation is further compounded by the scarcity of large-scale, high-quality training data for document parsing, which hinders models from acquiring layout-aware knowledge. To overcome these limitations, reinforcement learning (RL) presents a promising alternative, having demonstrated strong generalization capabilities in vision and multimodal tasks, where outcome-based rewards help models learn transferable representations (Huang et al., 2025; Liu et al., 2025a; Wang et al., 2025c). This raises a fundamental question: can reinforcement learning drive models toward generalizable layout parsing rules? Unfortunately, current RL approaches remain limited by coarse-grained, binary outcome rewards that fail to provide the fine-grained, layout-aware supervision necessary for modeling

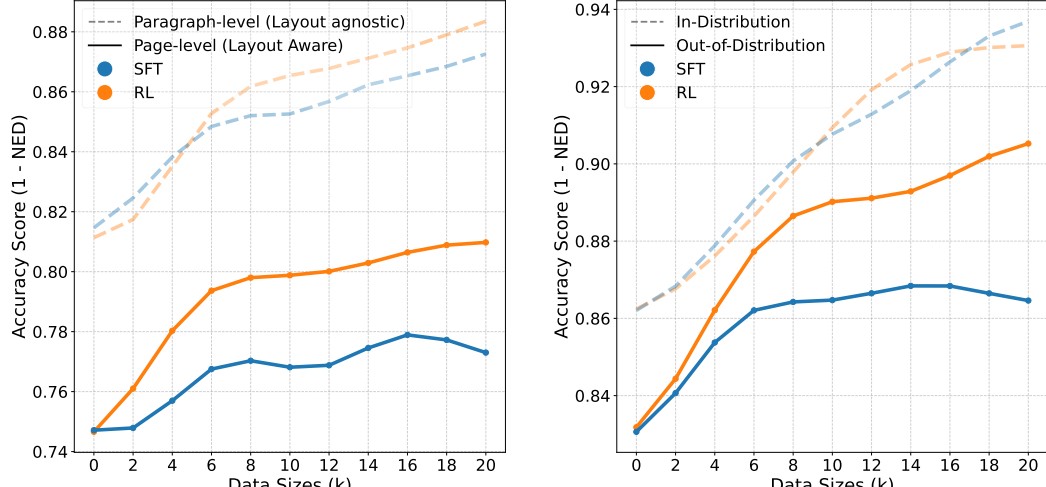

Figure 1: Comparison of document parsing performance on OmniDocBench under different training strategies as training data size increases. Left: Evaluation with two complementary metrics: (1) Paragraph-level accuracy (edit distance evaluation on element contents only), which assesses element-wise consistency within individual element contents, independent of inter-element reading order; and (2) Page-level accuracy (edit distance evaluation on element contents and reading order), which measures global document reconstruction quality by aligning predicted outputs (e.g., texts, tables, and formulas) with ground-truth sequences. Right: In-Distribution and Out-of-Distribution task performance measured by accuracy score (1 − NED). See detailed descriptions of the task in Section 4.3.

complex document layouts (Guo et al., 2025; Shao et al., 2024). Therefore, how to effectively apply RL to document parsing remains an underexplored yet critical challenge.

To address the challenge of effectively applying RL algorithms to document parsing tasks and the lack of large-scale, high-quality data in industry to support this training process, we propose LayoutRL, the first end-to-end reinforcement learning framework for layout-aware document parsing, and construct Infinity-Doc-400K, a large-scale dataset. Specifically, our approach does not rely on explicit reasoning processes, but rather treats the entire document parsing result as the final answer and guides model learning through carefully designed reward mechanisms. We introduce verifiable rewards (Shen et al., 2025; Liu et al., 2024a; Shao et al., 2024; Yaowei Zheng, 2025), which consist of Edit Distance Reward (Levenshtein et al., 1966) and Layout Parsing Reward signals, enforcing fine-grained alignment between predictions and ground-truth layouts and providing document-level supervision beyond token-level signals to encourage the model to learn transferable structural representations. Additionally, we construct a 400K-document corpus providing large-scale, high-quality supervision. It combines (1) high-fidelity synthetic scanned document parsing data—generated via HTML templates and browser rendering—and (2) expert-filtered real-world samples, pseudo-labeled through a cross-model agreement pipeline to capture genuine layout diversity. Built on this dataset, we train Infinity-Parser, an end-to-end VLM-based parser that directly outputs structured document representations.

To demonstrate the effectiveness of our approach, we conduct extensive experiments across multiple document parsing benchmarks. Results in Figure 1 show that while SFT achieves strong paragraph-level accuracy, its page-level performance struggles to improve as data size increases, reflecting its tendency to memorize surface patterns rather than model the hierarchical dependencies between document structures and elements. In contrast, our layout-aware reinforcement learning method significantly outperforms SFT in page-level accuracy, while maintaining competitive paragraph-level performance. These findings demonstrate that verifiable, layout-aware rewards enable models to move beyond simple token imitation, achieving consistent improvements in both local detail fidelity and global structural understanding, thereby establishing a more robust paradigm for document parsing. Moreover, our method also exhibits strong generalization to unseen task types in downstream applications. Finally, our method, Infinity-Parser, achieves new state-of-the-art results on four diverse benchmarks—OmniDocBench, olmOCR, PubTabNet, and FinTabNet—highlighting its effectiveness and strong cross-domain generalization.

We make the following contributions:

- We propose LayoutRL, a new reinforcement learning framework for end-to-end scanned document parsing, which explicitly trains models to be layout-aware by optimizing verifiable, multi-aspect rewards. Our multi-aspect reward design combines normalized edit distance, paragraph count accuracy, and reading order preservation, improving structural robustness.

- We introduce Infinity-Doc-400K, a large-scale dataset of 400,482 scanned documents that combines high-quality synthetic data with diverse real-world samples. The dataset features rich layout variations and comprehensive structural annotations, enabling robust training.

- We train a VLM based model, Infinity-Parser, which sets new state-of-the-art performance across English and Chinese benchmarks for OCR, table and formula extraction, and reading-order detection—demonstrating substantial gains in both structural fidelity and semantic accuracy over specialist pipelines and general-purpose vision-language models.

## 2 RELATED WORK

### 2.1 REINFORCEMENT LEARNING FOR LANGUAGE MODELS

Recent advancements in Large Language Models (LLMs) such as OpenAI's GPT series (OpenAI, 2024)), DeepSeek-R1 (Guo et al., 2025), and Gemini (Team et al., 2023) have highlighted the significant potential of Reinforcement Learning (RL) in enhancing their reasoning capabilities. This RL paradigm has been successfully extended to other domains demanding sophisticated reasoning, including code generation (Li et al., 2022; Zeng et al., 2025), autonomous tool utilization (Schick et al., 2023; Wang et al., 2025a), and information retrieval (Nakano et al., 2021). Similarly, RL has demonstrated its efficacy in the domain of Visual Language Models (VLMs), including precise object counting (Peng et al., 2025), nuanced visual perception (Liu et al., 2025b), and complex multimodal reasoning (e.g., VL-Rethinker (Wang et al., 2025b), Pixel Reasoner (Su et al., 2025), Vision-R1 (Huang et al., 2025)). These pioneering works have predominantly relied on binary outcome rewards to guide RL training. Complementary to these efforts, our work demonstrates the effectiveness of incorporating layout-aware and layout-based rewards for document parsing, offering a more granular and contextually relevant feedback mechanism.

### 2.2 VLM-BASED DOCUMENT PARSING

Recent advancements in document understanding and optical character recognition (OCR) have highlighted their importance as critical benchmarks for evaluating the perceptual capabilities of vision-language models (VLMs). By incorporating large-scale OCR corpora during pretraining, models such as GPT-4o (Achiam et al., 2023) and Qwen2-VL (Bai et al., 2024) have achieved competitive performance on document content extraction tasks. Building upon these foundations, the emergence of VLMs has further accelerated the progress of end-to-end document parsing, giving rise to a range of models such as Donut (Blecher et al., 2024), Nougat (Blecher et al., 2023), Kosmos-2.5 (Lv et al., 2024), Vary (Wei et al., 2025), mPLUG-DocOwl (Hu et al., 2024b), Fox (Liu et al., 2024b), and GOT (Wei et al., 2024). These models have continued to improve their understanding of visual layouts and textual content by leveraging advancements in visual encoders (Dosovitskiy et al., 2020), language decoders (Bai et al., 2024), and data construction pipelines. Despite the success of these VLM-based approaches in enabling end-to-end document parsing, they still face generalization challenges on downstream layout parsing tasks (Wang et al., 2024b). To address this issue, we propose leveraging reinforcement learning to provide a more effective training paradigm that better aligns with the demands of document parsing.

## 3 METHODOLOGY

In this section, we first introduce Infinity-Doc-400K, our large-scale multimodal dataset for end-to-end scanned document parsing. We then describe our rule-based multi-aspect reward framework, which integrates edit distance, paragraph count, and order criteria under a unified reinforcement learning objective optimized via Group Relative Policy Optimization (GRPO).

| Benchmark | Document Domain | Annotation Type | | | | | End-to-End Task | | | | Exactly Match |
|---|---|---|---|---|---|---|---|---|---|---|---|
| | | BBox | Text | Table | Formula | Attributes | OCR | TR | MFR | ROD | |
| *End-to-end Eval Benchmarks* | | | | | | | | | | | |
| Fox (Liu et al., 2024b) | 2 | ✔ | ✔ | | | | ✔ | | | | |
| Nougat (Blecher et al., 2024) | 1 | | ✔ | ✔ | ✔ | | ✔ | ✔ | ✔ | | |
| GOT OCR 2.0 (Wei et al., 2024) | 2 | | ✔ | ✔ | ✔ | | ✔ | ✔ | ✔ | | ✔ |
| OmniDocBench (Ouyang et al., 2024) | 9 | ✔ | ✔ | ✔ | ✔ | ✔ | ✔ | ✔ | ✔ | ✔ | ✔ |
| *End-to-end Train Dataset* | | | | | | | | | | | |
| DocStruct4M (Hu et al., 2024a) | - | | ✔ | | | | ✔ | | | | |
| olmOCR-mix (Poznanski et al., 2025) | - | | ✔ | ✔ | ✔ | | ✔ | ✔ | ✔ | ✔ | |
| **Infinity-Doc-400K** | 7 | ✔ | ✔ | ✔ | ✔ | ✔ | ✔ | ✔ | ✔ | ✔ | ✔ |

Table 1: A comparison between Infinity-Doc-400K and existing datasets. *BBox*: Bounding boxes. *Text*: Text in Unicode. *Table*: Table in LaTeX/HTML/Markdown. *Formula*: Formula in LaTeX. *Attributes*: Page- and BBox-Level Attributes. *OCR*: Optical Character Recognition; *TR*: Table Recognition; *MFR*: Math Formula Recognition; *ROD*: Reading Order Detection. *Multi-Type Doc*: Whether the dataset includes documents from multiple domains or categories.

## 3.1 INFINITY-DOC-400K AND GENERATION PIPELINES

We introduce Infinity-Doc-400K, a large-scale, multimodal dataset of 400,066 richly annotated documents for end-to-end scanned document parsing. Unlike prior benchmarks that target isolated subtasks (e.g., layout detection, OCR, or table recognition), Infinity-Doc-400K provides holistic supervision by pairing rendered scanned document pages with their ground-truth Markdown representations. This design enables training and evaluating models that directly translate visual inputs to layout outputs without relying on brittle, multi-stage pipelines. As shown in Table 1, compared to existing works, Infinity-Doc-400K not only significantly enhances task diversity but also substantially improves overall data quality through our proposed synthetic generation mechanism. More details on the data distribution and quality control are provided in the Data Details section of the Appendix.

To construct Infinity-Doc-400K, we design a dual-pipeline framework that integrates both synthetic and real-world document generation, as illustrated in Figure 2. This design addresses a critical limitation of traditional data construction pipelines, which often rely on weak supervision and pseudo-labeling from a single model applied to crawled, scanned documents. These pipelines frequently suffer from noisy, misaligned, or incomplete annotations, especially in complex layouts or multilingual content, thus hindering model performance and generalization. To overcome these issues, our dual-pipeline framework is motivated by the need to balance annotation quality and structural diversity. The synthetic branch provides highly accurate, clean, and precisely aligned annotations at scale, while the real-world branch introduces naturally occurring layout variability and semantic richness, which are essential for building models that generalize robustly in practical applications.

**Real-World Data** We develop a real-world data construction pipeline to capture the structural complexity and natural layout variability of documents across practical domains. We collect diverse scanned documents from sources such as financial reports, medical records, academic papers, books, magazines, and web pages, covering both dense and sparse content layouts. To generate structural annotations, we adopt a multi-expert strategy, where specialized models handle different structural elements, such as layout blocks, texts, formulas, and tables. For example, overall layouts are analyzed by a visual layout model (Huang et al., 2022), formula regions are processed by a dedicated formula recognition model (Wang et al., 2024a), and tables are parsed by a transformer-based table extractor (Blecher et al., 2024). A cross-validation mechanism is then applied to filter out inconsistencies by comparing the outputs of expert models and VLMs. Only regions with consistent predictions across models are retained as high-confidence pseudo-ground-truth annotations. This layout-aware filtering results in a rich and reliable dataset that reflects the complexity of real-world documents and supports robust document parsing model training.

**Synthetic Data** We design a synthetic data construction pipeline. We collect text and images from sources such as Wikipedia, web crawlers, and online corpora, and use Jinja (Nipkow, 2003) templates to inject sampled content into predefined single-, double-, or triple-column HTML layouts. These pages are rendered into scanned documents using a browser engine, followed by automated filtering to remove low-quality or overlapping images. Ground-truth annotations are extracted by

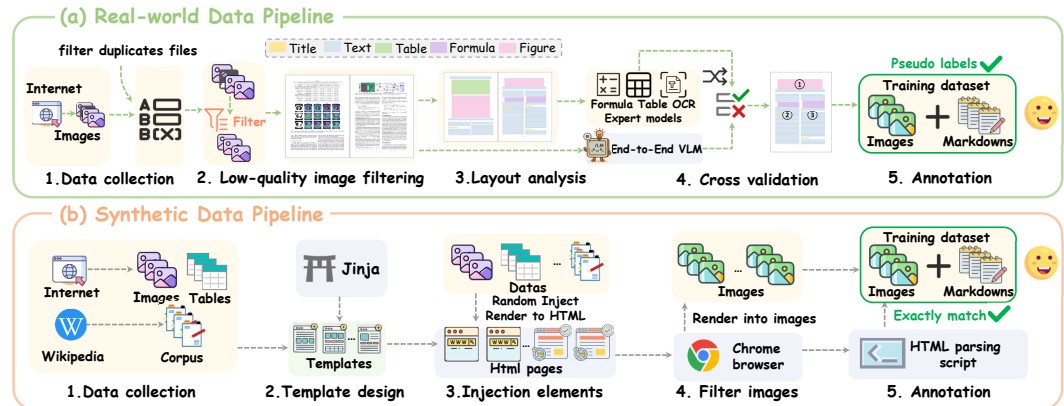

Figure 2: Data construction pipelines for document parsing. (a) Real-world pipelines enhance quality by combining multiple expert models and layout analysis, yielding better-aligned supervision through intersection and reading order reasoning. (b) Synthetic pipeline leverages structured HTML templates and browser rendering to generate clean, exactly-aligned scanned document parsing data, ensuring high-quality supervision for end-to-end parsing.

parsing the original HTML to produce aligned Markdown representations. This synthetic approach not only significantly reduces construction costs and ensures annotation accuracy and structural diversity, but more importantly, it addresses the longstanding issue of imprecise or inconsistent supervision commonly found in pseudo-labeled datasets, providing high-quality and well-aligned supervision for training end-to-end models.

## 3.2 RL with Layout-Aware Rewards

As illustrated in Figure 3, we employ an RL framework to directly optimize scanned document parsers, aiming to enhance both structural fidelity and semantic accuracy. Specifically, we utilize GRPO (Shao et al., 2024), which enables learning from rule-based reward signals without relying on absolute values. GRPO operates by generating a set of candidate Markdown outputs for each document and evaluating them using a *multi-aspect reward*, denoted as $R_{\text{Multi-Aspect}}$, which integrates multiple rule-based criteria into a unified supervisory signal. These raw rewards are then converted into relative advantage scores by comparing each candidate against others within the same group. This relative evaluation promotes training stability and encourages the selection of higher-quality outputs, eliminating the need for a learned value function or critic. Notably, our reinforcement learning approach avoids any explicit thinking or intermediate reasoning process; instead, all outputs are treated as final answers, with the model receiving verifiable rewards based on these outputs.

The multi-aspect reward $R_{\text{Multi-Aspect}}$ consists of three complementary components, each capturing a different aspect of parsing quality:

**Edit Distance Reward** ($R_{\text{dist}}$)  We define the edit distance reward based on the normalized Levenshtein distance $D(y, \hat{y})$ between the predicted output $\hat{y}$ and the reference output $y$:

$$R_{\text{dist}} = 1 - \frac{D(y, \hat{y})}{\max(N, M)} \tag{1}$$

where $N = |y|$ and $M = |\hat{y}|$ are the lengths of the reference and predicted sequences, respectively. The distance $D(y, \hat{y})$ measures the minimum number of single-character insertions, deletions, or substitutions required to convert $\hat{y}$ into $y$, thereby capturing both semantic and formatting discrepancies. This reward is bounded within $[0, 1]$, with higher values indicating better alignment between prediction and reference. Meanwhile, the reference output $y$ is synthesized through two data generation pipelines proposed in this work. It is produced with rigorous rule-based filtering and consistency validation, and serves as a high-quality surrogate for ground-truth annotations in evaluating the quality of the model output.

Figure 3: Overview of Infinity-Parser training framework. Our model is optimized via reinforcement finetuning with edit distance, layout, and order-based rewards.

**Count Reward** ($R_{\text{count}}$)  To encourage accurate paragraph segmentation, let $N_Y$ and $N_{\hat{Y}}$ be the numbers of reference and predicted paragraphs. We define:

$$R_{\text{count}} = 1 - \frac{|N_Y - N_{\hat{Y}}|}{N_Y} \tag{2}$$

which penalizes missing or spurious paragraphs.

**Order Reward** ($R_{\text{order}}$)  We measure sequence-level fidelity by counting pairwise inversions $D_{\text{order}}$ between reference and predicted paragraphs. with $\max_{\text{inv}} = N_Y(N_Y - 1)/2$, we set:

$$R_{\text{order}} = 1 - \frac{D_{\text{order}}}{\max_{\text{inv}}} \tag{3}$$

rewarding preservation of the original reading order.

The final multi-aspect reward is a weighted combination of these three components. Specifically, we begin by applying the Hungarian algorithm (Kuhn, 1955) to establish the optimal one-to-one matching between predicted and ground-truth segments, identifying both pairings and their relative order. Based on the matched segment count, we compute the count reward to reflect alignment in the number of segments. Using the relative sequence of matched pairs, we calculate the order reward to measure structural consistency. On top of these matchings, we compute the edit reward by averaging the edit similarities of each matched segment pair. Combining these terms yields the final reward:

$$R_{\text{Multi-Aspect}} = R_{\text{dist}} + R_{\text{count}} + R_{\text{order}} \tag{4}$$

This multi-aspect design balances content fidelity with structural correctness and order preservation, providing rich supervision for end-to-end document parsing.

## 4 EXPERIMENTS

We adopt Qwen2.5-VL-7B (Bai et al., 2025a) as the base model and apply the VeRL (Sheng et al., 2024) framework for reinforcement learning. Detailed implementation can be found in the Appendix.

### 4.1 MAIN RESULTS

We evaluate our method on several widely-used benchmarks for document understanding and OCR tasks. OmniDocBench (Ouyang et al., 2024) provides comprehensive evaluation across diverse document types using NED and TEDS metrics. For table recognition, we use PubTabNet with scientific tables and FinTabNet (Zheng et al., 2021) with financial documents. Additionally, we

| Methods | Overall$^{Edit}\downarrow$ | | Text$^{Edit}\downarrow$ | | Form.$^{Edit}\downarrow$ | | Table$^{TEDS}\uparrow$ | | Table$^{Edit}\downarrow$ | | Read Order$^{Edit}\downarrow$ | |
|---|---|---|---|---|---|---|---|---|---|---|---|---|
| | EN | ZH | EN | ZH | EN | ZH | EN | ZH | EN | ZH | EN | ZH |
| **Based on Pipeline Tools** | | | | | | | | | | | | |
| MinerU (Wang et al., 2024b) | 0.15 | 0.357 | **0.061** | 0.215 | **0.278** | 0.577 | 78.6 | 62.1 | 0.18 | 0.344 | 0.079 | 0.292 |
| Marker (Paruchuri, 2024) | 0.336 | 0.556 | 0.080 | 0.315 | 0.530 | 0.883 | 67.6 | 49.2 | 0.619 | 0.685 | 0.114 | 0.340 |
| Mathpix | 0.191 | 0.365 | 0.105 | 0.384 | 0.306 | 0.454 | 77.0 | 67.1 | 0.243 | 0.320 | 0.108 | 0.304 |
| Docling (Livathinos et al., 2025) | 0.589 | 0.909 | 0.416 | 0.987 | 0.999 | 1.000 | 61.3 | 25.0 | 0.627 | 0.810 | 0.313 | 0.837 |
| Pix2Text (Gurgurov & Morshnev, 2024) | 0.320 | 0.528 | 0.138 | 0.356 | 0.276 | 0.611 | 73.6 | 66.2 | 0.584 | 0.645 | 0.281 | 0.499 |
| Unstructured-0.17.2 | 0.586 | 0.716 | 0.198 | 0.481 | 0.999 | 1.000 | - | - | 1.000 | 0.998 | 0.145 | 0.387 |
| OpenParse-0.7.0 | 0.646 | 0.814 | 0.681 | 0.974 | 0.996 | 1.000 | 64.8 | 27.5 | 0.284 | 0.639 | 0.595 | 0.641 |
| **Based on Expert VLMs** | | | | | | | | | | | | |
| GOT-OCR (Wei et al., 2024) | 0.287 | 0.411 | 0.189 | 0.315 | 0.360 | 0.528 | 53.2 | 47.2 | 0.459 | 0.520 | 0.141 | 0.280 |
| Nougat (Blecher et al., 2024) | 0.452 | 0.973 | 0.365 | 0.998 | 0.488 | 0.941 | 39.9 | 0.0 | 0.572 | 1.000 | 0.382 | 0.954 |
| Mistral OCR | 0.268 | 0.439 | 0.072 | 0.325 | 0.318 | 0.495 | 75.8 | 63.6 | 0.600 | 0.650 | 0.083 | 0.284 |
| OLMOCR-sglang | 0.326 | 0.469 | 0.097 | 0.293 | 0.455 | 0.655 | 68.1 | 61.3 | 0.608 | 0.652 | 0.145 | 0.277 |
| SmolDocling-256M | 0.493 | 0.816 | 0.262 | 0.838 | 0.753 | 0.997 | 44.9 | 16.5 | 0.729 | 0.907 | 0.227 | 0.522 |
| **Based on General VLMs** | | | | | | | | | | | | |
| GPT-4o (Achiam et al., 2023) | 0.233 | 0.399 | 0.144 | 0.409 | 0.425 | 0.606 | 72.0 | 62.9 | 0.234 | 0.329 | 0.128 | 0.251 |
| Qwen2-VL-72B (Wang et al., 2024c) | 0.252 | 0.327 | 0.096 | 0.218 | 0.404 | 0.487 | 76.8 | 76.4 | 0.387 | 0.408 | 0.119 | 0.193 |
| InternVL2-76B (Chen et al., 2024) | 0.440 | 0.443 | 0.353 | 0.290 | 0.543 | 0.701 | 63.0 | 60.2 | 0.547 | 0.555 | 0.317 | 0.228 |
| Qwen2.5-VL-7B (Bai et al., 2025b) | 0.220 | 0.265 | 0.142 | 0.205 | 0.393 | 0.530 | 78.7 | 78.3 | 0.155 | 0.162 | 0.191 | 0.169 |
| InternVL3-8B (Zhu et al., 2025) | 0.426 | 0.385 | 0.315 | 0.345 | 0.714 | 0.729 | 59.0 | 71.5 | 0.352 | 0.211 | 0.324 | 0.257 |
| **Based on Reinforcement Learning** | | | | | | | | | | | | |
| Infinity-Parser-7B | **0.141** | **0.197** | 0.076 | **0.117** | 0.314 | **0.434** | **85.3** | **81.4** | **0.098** | **0.142** | **0.076** | **0.095** |

Table 2: Comprehensive evaluation of document parsing algorithms on OmniDocBench: performance metrics for text, formula, table, and reading order extraction, with overall scores derived from ground truth comparisons.

employ olmOCR-Bench (Poznanski et al., 2025) for fact-based OCR evaluation. We ensure that the test data for each benchmark undergoes rigorous text similarity filtering to prevent any overlap with the training data. Detailed descriptions of these benchmarks are provided in Appendix.

**Overall Evaluation on OmniDocBench** As shown in Table 2, pipeline-based methods such as MinerU (Wang et al., 2024b) and Mathpix achieve superior performance across individual sub-tasks including text recognition and formula recognition. Meanwhile, general-purpose vision-language models like Qwen2.5-VL-7B and GPT-4o also demonstrate competitive results. Notably, most methods perform better on English pages compared to Chinese pages, reflecting language-dependent challenges. In contrast, our proposed Infinity-Parser-7B achieves a more balanced performance across all sub-tasks and languages, setting new SOTA results with overall edit distances of 0.141 and 0.197. This highlights the advantage of reinforcement learning with multi-aspect rewards in enabling robust, end-to-end document parsing.

| Model | Overall | ArXiv | Old Scans Math | Tables | Old Scans | Headers&Footers | Multi Col. | Long-Tiny Text | Base |
|---|---|---|---|---|---|---|---|---|---|
| GOT OCR | 48.3 | 52.7 | 52.0 | 0.2 | 22.1 | 93.6 | 42.0 | 29.9 | 94.0 |
| Marker v1.6.2 | 59.4 | 24.3 | 22.1 | 69.8 | 24.3 | 87.1 | 71.0 | 76.9 | 99.5 |
| MinerU v1.3.10 | 61.5 | 75.4 | 47.4 | 60.9 | 17.3 | **96.6** | 59.0 | 39.1 | 96.6 |
| Mistral OCR API | 72.0 | 77.2 | 67.5 | 60.6 | 29.3 | 93.6 | 71.3 | 77.1 | 99.4 |
| GPT-4o (No Anchor) | 68.9 | 51.5 | 75.5 | 69.1 | 40.9 | 94.2 | 68.9 | 54.1 | 96.7 |
| GPT-4o (Anchored) | 69.9 | 53.5 | 74.5 | 70.0 | 40.7 | 93.8 | 69.3 | 60.6 | 96.8 |
| Gemini Flash 2 (No Anchor) | 57.8 | 32.1 | 56.3 | 61.4 | 27.8 | 48.0 | 58.7 | 84.4 | 94.0 |
| Gemini Flash 2 (Anchored) | 63.8 | 54.5 | 56.1 | 72.1 | 34.2 | 64.7 | 61.5 | 71.5 | 95.6 |
| Qwen 2 VL (No Anchor) | 31.5 | 19.7 | 31.7 | 24.2 | 17.1 | 88.9 | 8.3 | 6.8 | 55.5 |
| Qwen 2.5 VL (No Anchor) | 65.5 | 63.1 | 65.7 | 67.3 | 38.6 | 73.6 | 68.3 | 49.1 | 98.3 |
| olmOCR v0.1.68 (No Anchor) | 76.3 | 72.1 | 74.7 | 71.5 | 43.7 | 91.6 | 78.5 | 80.5 | 98.1 |
| olmOCR v0.1.68 (Anchored) | 77.4 | 75.6 | 75.1 | 70.2 | 44.5 | 93.4 | 79.4 | 81.7 | 99.0 |
| Infinity-Parser-7B | **82.5** | **84.4** | **83.8** | **85.0** | **47.9** | 88.7 | **84.2** | **86.4** | **99.8** |

Table 3: Performance comparison on the olmOCR (Poznanski et al., 2025) benchmark across multiple document domains and structural challenges. Higher is better.

**Document-level OCR Evaluation** Table 3 reports performance on the olmOCR-Bench benchmark, which evaluates document-level OCR across diverse layouts and domains. Infinity-Parser-7B achieves the highest overall score (82.5), followed closely by olmOCR v0.1.68 (Anchored) (77.4), both demonstrating strong performance in complex categories like multi-column layouts and scanned math content. The results highlight the effectiveness of anchored prompting, with anchored versions of models (e.g., GPT-4o, olmOCR) significantly outperforming their non-anchored counterparts—especially on tables and old scans. This underscores the importance of layout-aware extraction techniques. In contrast, traditional pipelines like Marker and GOT OCR lag behind in

structural accuracy, reinforcing the value of modern VLM-based approaches in high-fidelity PDF understanding.

**Table Recognition Evaluation**  To evaluate the model's generalization ability, we introduce task-specific test cases. In Table 4, we compare Infinity-Parser-7B with end-to-end table recognition models on PubTab-Net and FinTabNet using the TEDS metric, which evaluates both structure and content. We also report TEDS-S for structure-only assessment. The evaluation results for InternVL3, Qwen2.5-VL, and GPT-4o were generated through our standardized benchmarking pipeline. Infinity-Parser-7B achieves the highest TEDS-S and TEDS scores on both datasets.

| Model | PubTabNet | | FinTabNet | |
|---|---|---|---|---|
| | TEDS-S | TEDS | TEDS-S | TEDS |
| EDD | 89.9 | 88.3 | 90.6 | - |
| OmniParser | 90.45 | 88.83 | 91.55 | 89.75 |
| InternVL3-8B | 87.48 | 83.02 | 86.73 | 84.01 |
| InternVL3-78B | 89.63 | 82.11 | 92.51 | 89.21 |
| Qwen2.5-VL-7B | 86.78 | 81.60 | 87.46 | 82.58 |
| Qwen2.5-VL-72B | 87.91 | 84.39 | 87.13 | 82.90 |
| GPT-4o | 86.16 | 76.53 | 87.00 | 83.96 |
| **Infinity-Parser-7B** | **93.46** | **91.82** | **97.16** | **95.92** |

Table 4: Comparisons of end-to-end table recognition methods on PubTabNet and FinTabNet.

## 4.2 Ablation Study

We perform ablation experiments to evaluate the individual contributions of our three core design choices: (1) data quality verification and (2) multi-aspect rewards. We report all ablation results using two primary evaluation metrics: Overall$^{Edit}$ and Overall$^{Cat.}$. Overall$^{Edit}$ represents the average edit-based overall score across English and Chinese pages, as shown in Table 2. In contrast, Overall$^{Cat.}$ reflects the mean category-level performance across nine types of scanned document pages, following the same evaluation setting as Table 9 in the Appendix.

| Method | Edit Dist. | Count. | Order. | SFT | RL | Overall (EN)$^{Edit}$ ↓ | Overall (ZH)$^{Edit}$ ↓ | Overall $^{Cat.}$ ↓ |
|---|---|---|---|---|---|---|---|---|
| Zero Shot | - | - | - | - | - | 0.220 | 0.265 | 0.183 |
| SFT | - | - | - | 43K | - | 0.198 | 0.261 | 0.159 |
| Zero + RL | ✔ | - | - | - | 43K | 0.169 | 0.224 | 0.156 |
| Zero + RL | ✔ | ✔ | - | - | 43K | 0.159 | 0.200 | 0.112 |
| Zero + RL | ✔ | ✔ | ✔ | - | 43K | 0.141 | 0.197 | 0.104 |
| SFT + RL | ✔ | ✔ | ✔ | 43K | 43K | 0.163 | 0.195 | 0.092 |

Table 5: Results under different reward designs.

Effect of Multi-Aspect Rewards. Table 5 demonstrates that reinforcement learning can outperform supervised fine-tuning when appropriate reward designs are applied. Compared to the SFT baseline (0.198 / 0.261 / 0.159), the RL method with distance-based reward ($Zero + R_{dist}$) achieves better Overall$^{Edit}$ (EN: 0.169 vs. 0.198, ZH: 0.224 vs. 0.261) while maintaining a comparable Overall$^{Cat.}$ (0.156 vs. 0.159). Incorporating additional count and order rewards further improves structural consistency: $Zero + R_{dist} + R_{count}$ achieves 0.159 / 0.200 / 0.112, and $Zero + R_{dist} + R_{count} + R_{order}$ achieves 0.141 / 0.197 / 0.104. Meanwhile, when reinforcement learning is combined with supervised fine-tuning ($SFT + RL$), the model does not exhibit further significant improvements. This observation is consistent with existing studies (Guo et al., 2025), which suggest that given the backbone model's inherent capabilities, SFT may be unnecessary for downstream RL training in certain scenarios. These results further demonstrate that reinforcement learning, when equipped with structural supervisory signals, enables the model to better align with task-specific objectives.

## 4.3 Further Analysis of LayoutRL

**Training Stability Across Task Types.**  As shown in Figure 4, we compare SFT and Layout-Aware RL on four OmniDocBench sub-tasks: table recognition, text recognition, formula parsing, and reading order prediction. Across all tasks, RL consistently achieves better performance, yielding higher scores on TEDS and lower errors on NED. More importantly, the RL curves exhibit smoother trajectories with stable improvements over training, while SFT shows large fluctuations and even performance regressions at certain stages. These results indicate that layout-aware rewards not only improve final accuracy but also enhance training stability throughout optimization.

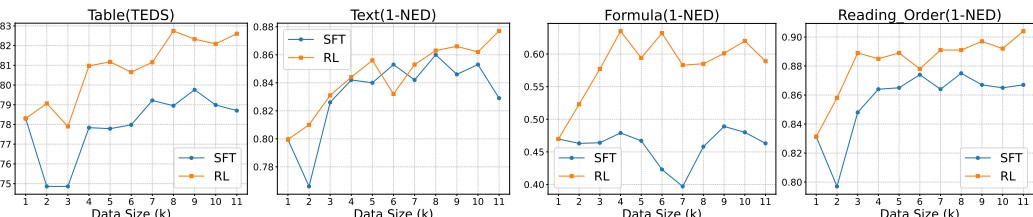

Figure 4: Performance comparison of SFT and Layout-Aware RL on OmniDocBench sub-tasks.

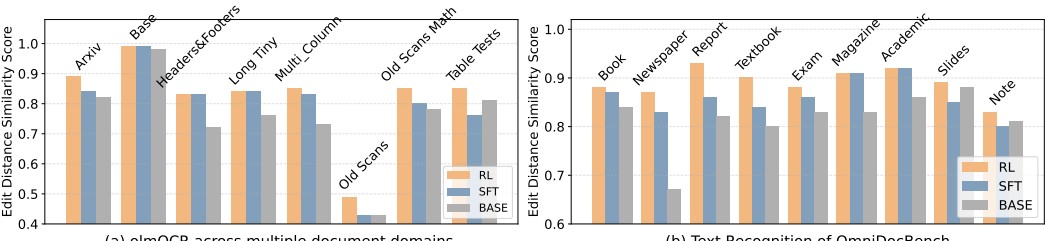

Figure 5: Comparison of model performance on different document parsering tasks.

**Robustness Across Diverse Document Tasks.**    Figure 5 compares RL, SFT, and Zero-Shot (Base) across diverse document parsing tasks. On olmOCR (left), RL consistently achieves higher Levenshtein distance similarity score, especially on challenging cases like old scans and table tasks. At the same time, SFT offers moderate gains over Zero-Shot but remains behind RL. On OmniDocBench (right), RL also outperforms the other methods across most document types, showing notable improvements on books, reports, and academic texts. Overall, RL demonstrates greater robustness and better generalization in both structural parsing on olmOCR and text recognition on omnidocbench.

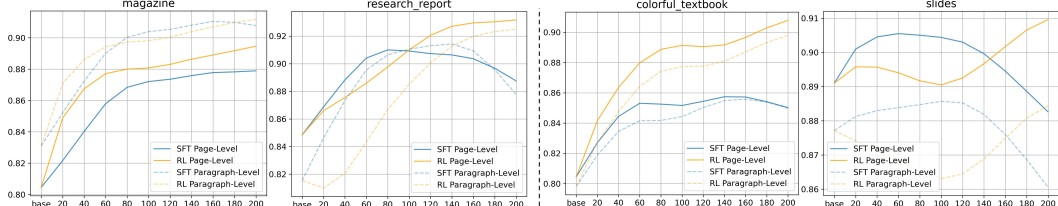

Figure 6: Generalization comparison of SFT and Layout-Aware RL. X-axis represents training steps. Y-axis represents 1-NED scores.

**Analysis of Generalization**    The Figure 6 compares document parsing performance across different training steps. The left two plots (magazine, research report) correspond to In-Distribution settings, where training and evaluation domains are aligned, while the right two plots (colorful textbook, slides) correspond to OOD settings, where evaluation involves unseen document types. In the in-distribution case, SFT achieves stable paragraph-level accuracy but its page-level performance tends to plateau, reflecting reliance on surface patterns. By contrast, RL continues to improve with data scale, achieving notable gains in page-level accuracy. In the OOD case, SFT performance degrades more severely, while RL maintains robustness and shows stronger improvements, highlighting its ability to capture global structural dependencies and generalize across distributions.

## 5    CONCLUSION

We introduced LayoutRL, an end-to-end reinforcement learning framework that explicitly incorporates layout awareness into document parsing through verifiable, multi-aspect rewards. To support this training, we built Infinity-Doc-400K, a large-scale dataset combining synthetic and real-world documents with diverse layouts, and trained Infinity-Parser, a VLM-based parser. Experiments on OmniDocBench, olmOCR-Bench, PubTabNet, and FinTabNet show that our approach achieves state-of-the-art performance across languages and document types, outperforming both specialized pipelines and general-purpose VLMs. Beyond accuracy, LayoutRL improves training stability and demonstrates robustness across diverse document tasks, highlighting reinforcement learning as a promising direction for robust and transferable document intelligence.

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
