## A    USE OF LARGE LANGUAGE MODELS

We used a large language model (GPT4V) as an assistive tool during the preparation of this paper. Specifically, it was employed for language polishing, grammar refinement, and providing alternative phrasings to improve readability. The model was not involved in generating research ideas, designing experiments, analyzing results, or drawing conclusions. All substantive contributions, including research conception, methodology, data analysis, and interpretation, were performed solely by the authors, who take full responsibility for the content.

## B    ETHICS STATEMENT

This work does not involve human subjects, personal data, or sensitive user information. All experiments are conducted on publicly available datasets or synthetic data generated via HTML rendering. The newly introduced Infinity-Doc-400K dataset combines automatically generated synthetic documents with real-world samples that are pseudo-labeled through cross-model agreement and manually filtered to ensure quality. We have taken care to remove potentially harmful or inappropriate content, and the dataset will be released strictly for research purposes under an academic license. We believe our contributions do not pose risks of discrimination, bias, or privacy violations, and instead aim to advance the robustness and reliability of document parsing technologies for broad scientific and practical use.

## C    REPRODUCIBILITY STATEMENT

We are committed to ensuring the reproducibility of our results. To this end, we will release the full Infinity-Doc-400K dataset, the implementation of LayoutRL with verifiable reward functions, and the pretrained Infinity-Parser model. Detailed training configurations, hyperparameters, and evaluation protocols are provided in the appendix and supplementary materials. Our experiments are conducted on widely used benchmarks, including OmniDocBench, olmOCR-Bench, PubTabNet, and FinTabNet, ensuring comparability with prior work. We will also provide scripts for preprocessing, training, and evaluation to facilitate reproducibility and further research by the community.

## D    TRAINING DETAILS

### D.1    TRAINING CONTEXT LENGTH DISTRIBUTION

Our model was trained with a maximum context length of 8K tokens. To provide a clearer picture of the training data, we report detailed statistics of the context length distribution in Table 6 and Table 7. The average context length is 1,765 tokens, with a maximum of 31,147 tokens. More than 73% of the samples fall within the [512, 4K) range. For sequences exceeding the 8K limit, we applied a left-truncation strategy to retain the semantically more relevant content at the end of the sequence.

| Metric | Min | Max | Average | Median | Std |
|--------|-----|-----|---------|--------|-----|
| Value | 17 | 31,147 | 1,765 | 1,127 | 1,692 |

Table 6: Summary statistics of training context length.

| Context Length (tokens) | [1,256) | [256,512) | [512,1K) | [1K,2K) | [2K,4K) | [4K,8K) | [8K,16K) | ≥16K |
|-------------------------|---------|-----------|----------|---------|---------|---------|----------|------|
| Frequency (count) | 25,125 | 40,680 | 119,041 | 102,757 | 70,554 | 39,780 | 2,482 | 63 |
| Distribution (%) | 6.27 | 10.16 | 29.72 | 25.66 | 17.62 | 9.93 | 0.62 | 0.02 |

Table 7: Distribution of training samples across different context length intervals.

# E  DATA DETAILS

As illustrated in Figure 7, the dataset spans seven diverse document domains, making it one of the most richly annotated and structurally varied resources to date. Each domain is represented by two sample pages, highlighting the broad variability in layout design, content structure, and semantic density. For instance, Medical Reports typically contain structured tables with clinical measurements and diagnostic notes. Synthetic Documents are algorithmically generated to replicate real-world formats, providing layout diversity for training robust parsers. Financial Reports feature dense tables and formal accounting records, while Academic Papers often follow two-column layouts with references, equations, and figures. Books combine narrative content with visual illustrations, and Magazines blend images and stylized text for reader engagement. Finally, Web Pages, when saved as PDFs, preserve HTML-based structures that integrate tables, lists, and dynamic elements. This visual taxonomy exemplifies the structural and semantic diversity present in real-world documents, highlighting the core challenge faced by document AI systems: reliably parsing heterogeneous layouts and extracting structured information across a wide variety of formats.

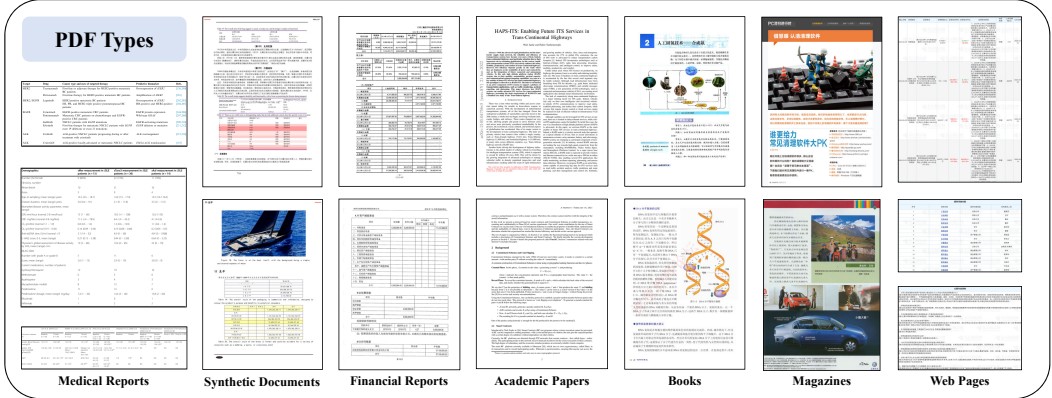

Figure 7: This figure illustrates a diverse collection of PDF document types commonly encountered in Infinity-Doc-55K, grouped into seven categories: Medical Reports, Synthetic Documents, Financial Reports, Academic Papers, Books, Magazines, and Web Pages.

Table 8 provides an overview of the document types included in the Infinity-Doc-400K dataset, detailing the composition across both real-world and synthetic sources. The real-world portion consists of 331K documents spanning six domains: financial reports, medical reports, academic papers, books, magazines, and web pages. These documents were collected from the web and annotated using a pseudo-labeling pipeline based on expert model agreement. While this approach enables large-scale data acquisition, the resulting label quality is relatively low due to occasional inconsistencies across models. Additionally, real-world data collection incurs a high cost, especially in terms of manual filtering, formatting normalization, and layout validation.

| Data Source | Document Types | Size | Annotation Method | Cost |
|---|---|---|---|---|
| Real-World Doc | Financial Reports | 58.0K | Web + Pseudo-Label | High |
| Real-World Doc | Medical Reports | 5.0K | Web + Pseudo-Label | High |
| Real-World Doc | Academic Papers | 71.7K | Web + Pseudo-Label | High |
| Real-World Doc | Books | 11.3K | Web + Pseudo-Label | High |
| Real-World Doc | Magazines | 180.0K | Web + Pseudo-Label | High |
| Real-World Doc | Web Pages | 5.0K | Web + Pseudo-Label | High |
| Synthetic | Synthetic Documents | 69.0K | CC3M + Web + Wiki | Low |

Table 8: Overview of document types in the Infinity-Doc-400K dataset, including data source, document type, annotation method, and collection cost.

**Quality Control Measures**  To ensure reliable annotations at scale, we designed a hybrid quality control strategy for Infinity-Doc-400K. First, three domain experts with doctoral degrees in docu-

ment analysis manually inspected about 5% of the data. Their feedback not only identified potential errors but also served as a quality anchor for evaluating annotation consistency. Guided by these inspections, we iteratively refined the screening rules no fewer than five times, continuously improving the labeling pipeline. Finally, to achieve scalability, we employed a model-based cross-verification mechanism: multiple models generated annotations for real-world samples, with high-consistency outputs retained and inconsistent cases fed back into further rule refinement. This layered framework—anchored by expert inspection, strengthened through iterative rule optimization, and scaled via model cross-checking—effectively balances annotation reliability with dataset scalability.

## F  BENCHMARKS DETAILS

**OmniDocBench (Ouyang et al., 2024)** We conduct evaluation on OmniDocBench, a comprehensive benchmark that covers diverse document types and content modalities. To assess parsing performance across different structural elements, we employ two primary evaluation metrics: Normalized Edit Distance (NED), which measures the minimum edit operations required to transform one string into another normalized by the target string length, and Tree Edit Distance-based Similarity (TEDS), which captures structural similarities by comparing tree representations of HTML tables. These metrics are applied to different subtasks: NED is used to evaluate pure text, formula transcription, and reading order; TEDS combined with NED is used to evaluate both structural and content accuracy of table parsing.

**PubTabNet (Zhong et al., 2020)** A widely used benchmark for table recognition, containing 500,777 training and 9,115 validation images with diverse scientific table structures. Evaluation is conducted on the validation set.

**FinTabNet (Zheng et al., 2021)** Focused on financial documents, this dataset includes 112,000 single-page scanned documents, with 92,000 cropped training images and 10,656 for testing. It features dense layouts and detailed annotations for both structure and content evaluation.

**olmOCR-Bench (Poznanski et al., 2025)** This is a benchmark developed to automatically and reliably evaluate document-level OCR performance across a wide range of tools. Unlike traditional evaluation metrics such as edit distance—which may penalize valid variations or fail to capture critical semantic errors—olmOCR-Bench focuses on verifying simple, unambiguous, and machine-checkable "facts" about each document page, similar to unit tests. For instance, it checks whether a specific sentence appears exactly in the OCR output. The benchmark operates directly on single-page PDFs to preserve digital metadata, which can be beneficial for certain OCR systems, and to maintain the integrity of the original document format. Designed for flexibility, olmOCR-Bench supports outputs in Markdown or plain text, allowing for seamless evaluation of both open-source and custom OCR pipelines.

## G  IMPLEMENTATION DETAILS

We fine-tune the Qwen2.5-VL-7B model using GRPO within a distributed training setup based on Verl Sheng et al. (2024); Yaowei Zheng (2025), utilizing 8 A100 GPUs (80GB). Throughout our experiments, we set the KL coefficient $\beta = 1.0 \times 10^{-2}$. And for each problem instance, we sample 8 responses, each with a maximum length of 8192 tokens and a temperature of 1.0. Both the rollout batch size and the global batch size are set to 128. The actor model is updated using the AdamW optimizer with parameters $(\beta_1 = 0.9, \beta_2 = 0.99)$ and a learning rate $1.0 \times 10^{-6}$. The model is trained for 1.0 epoch for all experiments. Due to limited computational resources, we randomly sampled 43K documents from the 400K corpus for training. In our main results, we directly performed reinforcement learning on the base model using the 43K subset.

## H  MORE RESULTS

**Diverse Page Types Evaluation** To further investigate model behavior across diverse document types, we evaluated text recognition performance on nine distinct page categories. As shown in Table 9, pipeline-based systems such as MinerU Wang et al. (2024b) and Mathpix achieved strong

results on structured formats like academic papers and financial reports. General-purpose vision-language models (VLMs) demonstrated better generalization on less formal page types, including presentation slides and handwritten notes. However, for challenging formats such as newspapers, most VLMs underperformed, while pipeline tools maintained relatively lower error rates. Notably, our proposed Infinity-Parser-7B achieved consistently low edit distances across all document types, outperforming both pipeline-based systems and general-purpose VLMs in overall accuracy. This highlights the robustness and adaptability of our reinforcement learning approach across diverse and complex document layouts.

| Models | Book | Slides | Financial Report | Textbook | Exam Paper | Magazine | Academic Papers | Notes | Newspaper | Overall ↓ |
|---|---|---|---|---|---|---|---|---|---|---|
| *Based on Pipeline Tools* | | | | | | | | | | |
| MinerU | **0.055** | 0.124 | **0.033** | 0.102 | 0.159 | **0.072** | **0.025** | 0.984 | 0.171 | 0.206 |
| Marker | 0.074 | 0.34 | 0.089 | 0.319 | 0.452 | 0.153 | 0.059 | 0.651 | 0.192 | 0.274 |
| Mathpix | 0.131 | 0.22 | 0.202 | 0.216 | 0.278 | 0.147 | 0.091 | 0.634 | 0.69 | 0.3 |
| *Based on Expert VLMs* | | | | | | | | | | |
| GOT-OCR | 0.111 | 0.222 | 0.067 | 0.132 | 0.204 | 0.198 | 0.179 | 0.388 | 0.771 | 0.267 |
| Nougat | 0.734 | 0.958 | 1.000 | 0.820 | 0.930 | 0.83 | 0.214 | 0.991 | 0.871 | 0.806 |
| *Based on General VLMs* | | | | | | | | | | |
| GPT-4o | 0.157 | 0.163 | 0.348 | 0.187 | 0.281 | 0.173 | 0.146 | 0.607 | 0.751 | 0.316 |
| Qwen2-VL-72B | 0.096 | **0.061** | 0.047 | 0.149 | 0.195 | 0.071 | 0.085 | 0.168 | 0.676 | 0.179 |
| InternVL2-76B | 0.216 | 0.098 | 0.162 | 0.184 | 0.247 | 0.150 | 0.419 | 0.226 | 0.903 | 0.3 |
| Qwen2.5-VL-7B | 0.222 | 0.131 | 0.194 | 0.268 | 0.203 | 0.230 | 0.195 | 0.249 | 0.394 | 0.230 |
| InternVL3-8B | 0.311 | 0.233 | 0.320 | 0.222 | 0.238 | 0.157 | 0.438 | 0.268 | 0.726 | 0.328 |
| *Based on Reinforcement Learning* | | | | | | | | | | |
| Infinity-Parser-7B | 0.112 | 0.107 | 0.070 | **0.093** | **0.082** | 0.082 | 0.087 | **0.141** | **0.153** | **0.104** |

Table 9: End-to-end text recognition performance on OmniDocBench: evaluation using edit distance across 9 PDF page types. We compare with Mathpix, MinerU (Wang et al., 2024b), Marker (Paruchuri, 2024), GOT-OCR (Wei et al., 2024), Nougat (Blecher et al., 2024), GPT-4o (**?**), Qwen2-VL-72B (Wang et al., 2024c), InternVL2-76B (Chen et al., 2024), Qwen2.5-VL-7B (Bai et al., 2025b), InternVL3-8B (Zhu et al., 2025).

Table 10 summarizes the performance of various models on the OmniDocBench table subset, evaluated along three dimensions: language diversity, table frame types, and special layout conditions. Notably, Infinity-Parser-7B achieves the best overall performance with an impressive score of 86.4, outperforming all other models across most individual metrics. It leads in nearly every category, including mixed-language settings (94.8), complex frame layouts (e.g., omission and three-line formats), and challenging special situations such as merged cells, formulas, and rotations. This demonstrates its strong generalization ability and robustness across diverse and noisy table formats.

| Model | Language | | | Table Frame Type | | | | Special Situation | | | | Overall ↑ |
|---|---|---|---|---|---|---|---|---|---|---|---|---|
| | EN | ZH | Mixed | Full | Omission | Three | Zero | Merge Cell(+/-) | Formula(+/-) | Colorful(+/-) | Rotate(+/-) | |
| PaddleOCR (Li et al., 2022a) | 76.8 | 71.8 | 80.1 | 67.9 | 74.3 | 81.1 | 74.5 | 70.6/75.2 | 71.3/74.1 | 72.7/74.0 | 23.3/74.6 | 73.6 |
| RapidTable (RapidAI, 2023) | 80.0 | 83.2 | 91.2 | 83.0 | 79.7 | 83.4 | 78.4 | 77.1/85.4 | 76.7/83.9 | 77.6/84.9 | 25.2/83.7 | 82.5 |
| StructEqTable (Zhou et al., 2024) | 72.8 | 75.9 | 83.4 | 72.9 | 76.2 | 76.9 | 88.0 | 64.5/81.0 | 69.2/76.6 | 72.8/76.4 | 30.5/76.2 | 75.8 |
| GOT-OCR (Wei et al., 2024) | 72.2 | 75.5 | 85.4 | 73.1 | 72.7 | 78.2 | 75.7 | 65.0/80.2 | 64.3/77.3 | 70.8/76.9 | 8.5/76.3 | 74.9 |
| Qwen2-VL-7B (Wang et al., 2024c) | 70.2 | 70.7 | 82.4 | 70.2 | 62.8 | 74.5 | 80.3 | 60.8/76.5 | 63.8/72.6 | 71.4/70.8 | 20.0/72.1 | 71.0 |
| InternVL2-8B (Chen et al., 2024) | 70.9 | 71.5 | 77.4 | 69.5 | 69.2 | 74.8 | 75.8 | 58.7/78.4 | 62.4/73.6 | 68.2/73.1 | 20.4/72.6 | 71.5 |
| Qwen2.5-VL-7B (Wang et al., 2024c) | 87.4 | 80.7 | 93.5 | 86.4 | 85.1 | 84.1 | 88.7 | 77.5/89.8 | 82.1/87.2 | 77.1/87.5 | 56.5/86.0 | 85.5 |
| InternVL3-8B (Zhu et al., 2025) | 79.5 | 86.0 | 91.7 | 85.5 | 80.7 | 83.9 | 85.9 | 71.9/90.9 | 74.0/86.7 | 82.1/85.3 | 12.6/85.5 | 84.3 |
| **Infinity-Parser-7B** | 84.7 | **86.7** | **94.8** | 85.5 | **86.5** | **87.4** | **89.4** | **78.6/90.7** | 81.9/**87.5** | **83.2/88.0** | 68.8/86.7 | **86.4** |

Table 10: Component-level Table Recognition evaluation on OmniDocBench table subset. *(+/-)* means *with/without* special situation.

# I   PROMPT STRATEGY FOR PARSING TASKS.

**Prompt Template** summarizes the prompt designs for two key parsing tasks: document parsing and table parsing. For document parsing, the prompts instruct the model to recognize visual regions and convert their contents into structured Markdown. This design ensures consistent region-level extraction across documents with diverse layouts.

For table parsing, although the prompts are phrased differently, they share the same objective: transforming table content from images into HTML. This diversity encourages the model to general-

---

**Prompt Template**

**Document Parsing:** You are an AI assistant specialized in converting PDF images to Markdown format. Please follow these instructions for the conversion:

1. Text Processing:
- Accurately recognize all text content in the PDF image without guessing or inferring.
- Convert the recognized text into Markdown format.
- Maintain the original document structure, including headings, paragraphs, lists, etc.
2. Mathematical Formula Processing:
- Convert all mathematical formulas to LaTeX format.
- Enclose inline formulas with $ $. For example: This is an inline formula $E = mc^2$
- Enclose block formulas with $$ $$. For example:

$$\frac{-b \pm \sqrt{b^2 - 4ac}}{2a}$$

3. Table Processing:
- Convert tables to Markdown format.
4. Figure Handling:
- Ignore figures content in the PDF image. Do not attempt to describe or convert images.
5. Output Format:
- Ensure the output Markdown document has a clear structure with appropriate line breaks between elements.
- For complex layouts, try to maintain the original document's structure and format as closely as possible.
Please strictly follow these guidelines to ensure accuracy and consistency in the conversion. Your task is to accurately convert the content of the PDF image into Markdown format without adding any extra explanations or comments.

**Table Parsing:**

1. Please encode the table from the image into HTML format.

2. Render the table in the image as HTML code, please.

3. Please transform the table from the image into HTML format.

4. Convert the image's table data into the HTML structure.

5. Transform the image's table into the HTML format, please.

6. Convert the table found in the image into HTML format.

**Example Input:** A PDF with headings, paragraphs, and a table.
**Example Output:** Markdown reconstruction with proper hierarchy.

---

ize across variations in phrasing and reduces overfitting to a single instruction template. Notably, HTML is used here to match the evaluation format, but the resulting outputs can be easily converted to Markdown if needed for downstream use.

## J  CASE ANALYSIS

Figure 8 illustrates a progressive improvement in Markdown generation quality across different training strategies. The zero-shot model fails to capture key structural elements, omitting titles and producing redundant or incomplete content. With SFT, the model better identifies section headers and general layout but still suffers from symbol-level errors and repeated outputs. In contrast, the layout-aware RL model demonstrates the most accurate and coherent result, successfully preserving the document hierarchy and eliminating redundancy. This highlights the effectiveness of layout-aware rewards in guiding the model toward semantically and structurally faithful document parsing.

Infinity-Parser exhibits consistent improvements across a wide spectrum of document types, including academic papers, books, colorful textbooks, exam papers, magazines, government notices,

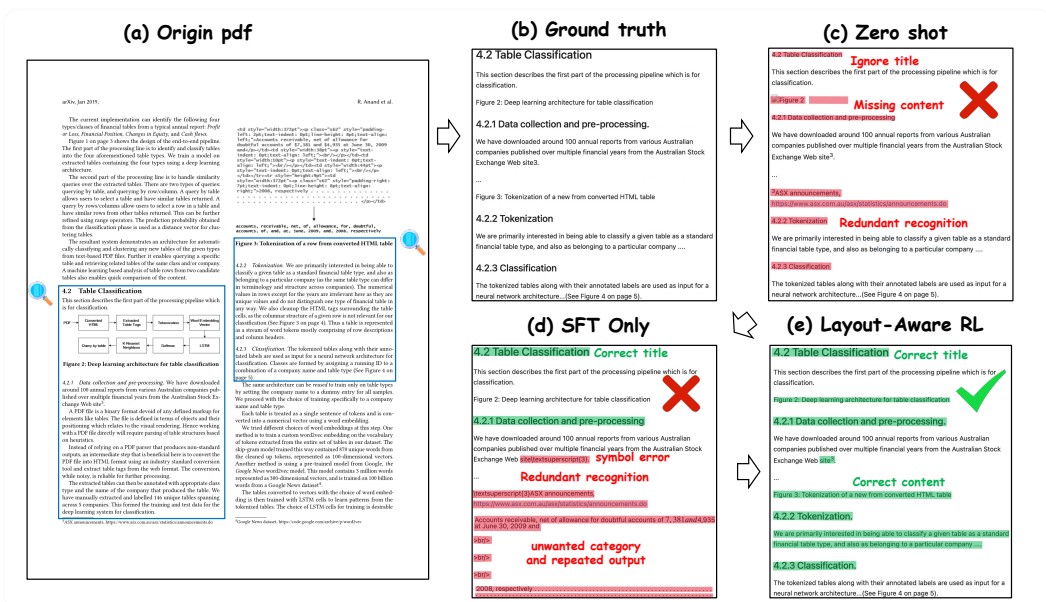

Figure 8: Comparison of four Markdown generation results on a single case, illustrating progressive improvement from direct inference to full reward integration.

newspaper articles, and PowerPoint-style slides. These gains are reflected in structural parsing, title and content recognition, formatting accuracy, and robustness to diverse visual layouts. As shown in Figures 9 through 14, we provide detailed visual comparisons with existing models, where Infinity-Parser consistently achieves superior results. These findings underscore the effectiveness and generalizability of our layout-aware RL approach across complex, real-world PDF formats.

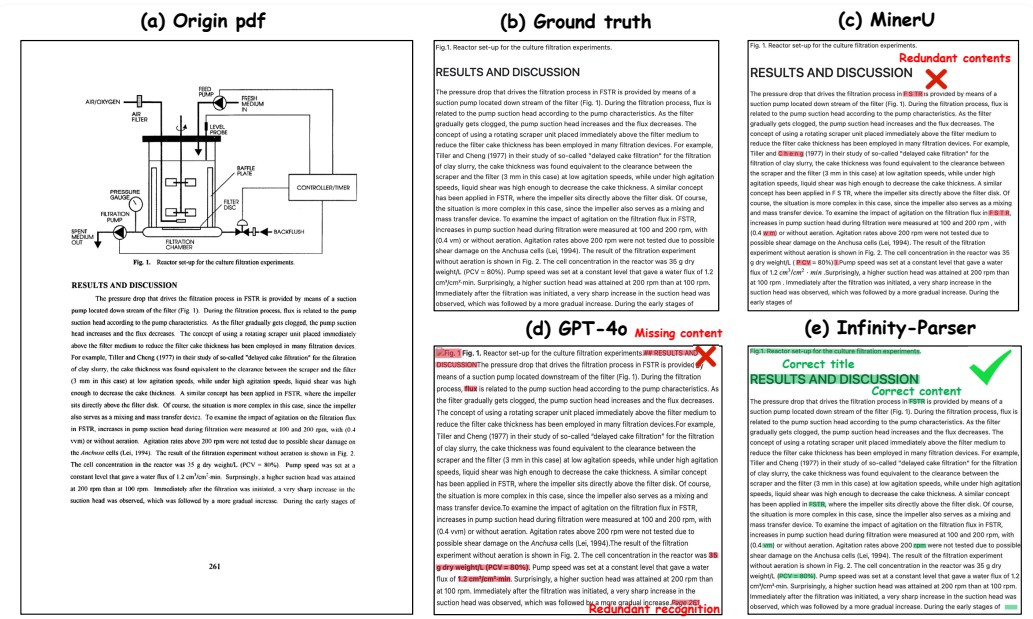

Figure 9: Comparison of Markdown extraction from academic literature using different models. The figure shows the original PDF (a), ground truth annotations (b), and extraction results from three models: MinerU, GPT-4o, and Infinity-Parser (c–e). Infinity-Parser produces the most accurate output, correctly identifying titles and content while avoiding redundancy and omissions.

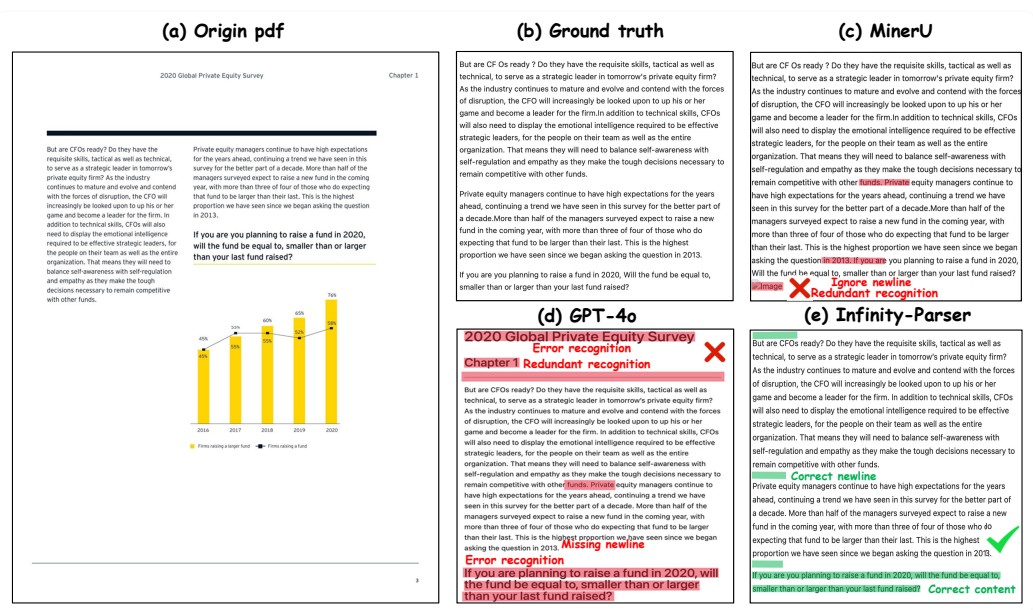

Figure 10: Comparison of Markdown extraction from a book-style PDF using different models. This figure displays the original document (a), human-annotated ground truth (b), and the extraction results from MinerU, GPT-4o, and Infinity-Parser (c–e). GPT-4o introduces multiple errors, such as incorrect headings, redundancy, and missing lines. MinerU retains unnecessary line breaks and repeated text. In contrast, Infinity-Parser correctly identifies the content structure, including titles and paragraphs, producing clean and accurate output.

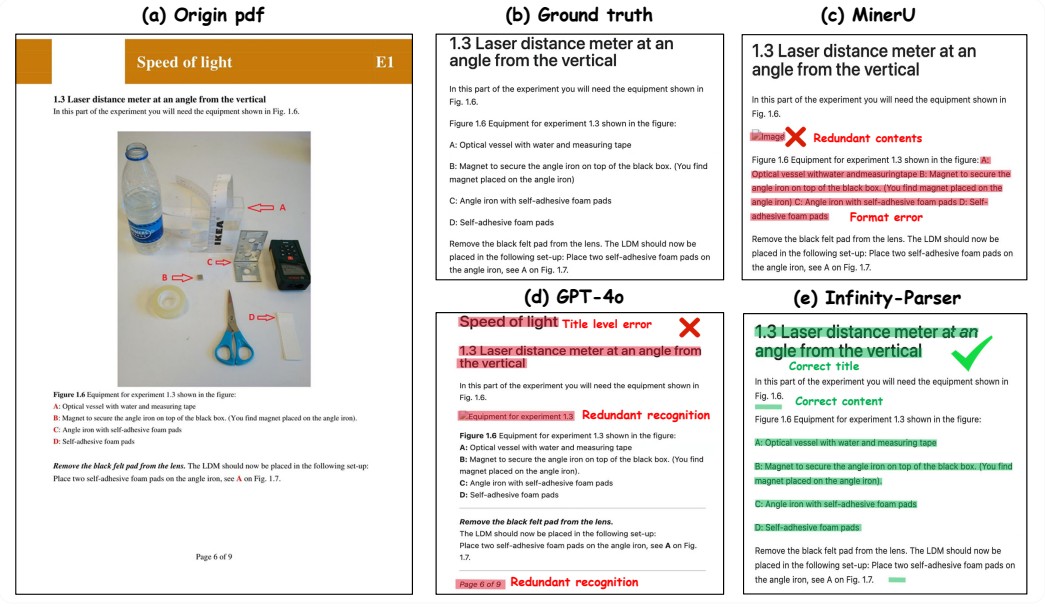

Figure 11: Comparison of Markdown extraction from an exam-style PDF. The figure presents the original exam document (a), ground truth annotations (b), and the extraction results from MinerU, GPT-4o, and Infinity-Parser (c–e). GPT-4o and MinerU both introduce redundant text and formatting errors, such as incorrect title levels and misplaced content. In contrast, Infinity-Parser accurately captures the heading hierarchy and structured list format, faithfully reproducing the content as intended in the original layout.

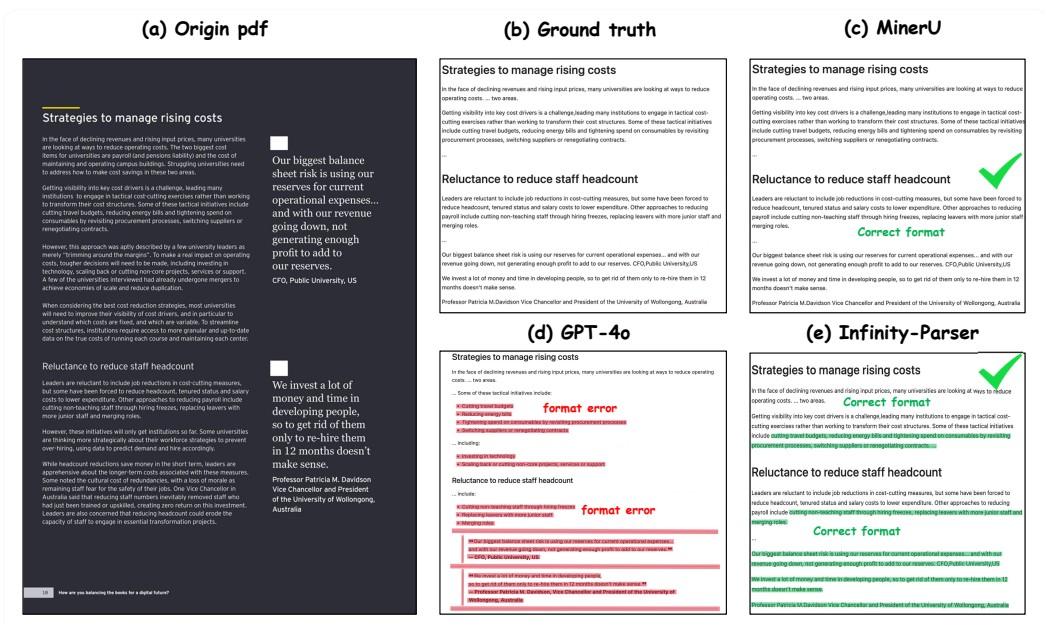

Figure 12: Comparison of Markdown extraction from a magazine-style PDF. The figure shows the original visually-rich page (a), ground truth annotations (b), and results from MinerU, GPT-4o, and Infinity-Parser (c–e). Due to the complex layout and dark background, GPT-4o suffers from significant formatting errors. Infinity-Parser accurately preserves the structural hierarchy and formatting, demonstrating robustness in handling stylized layouts.

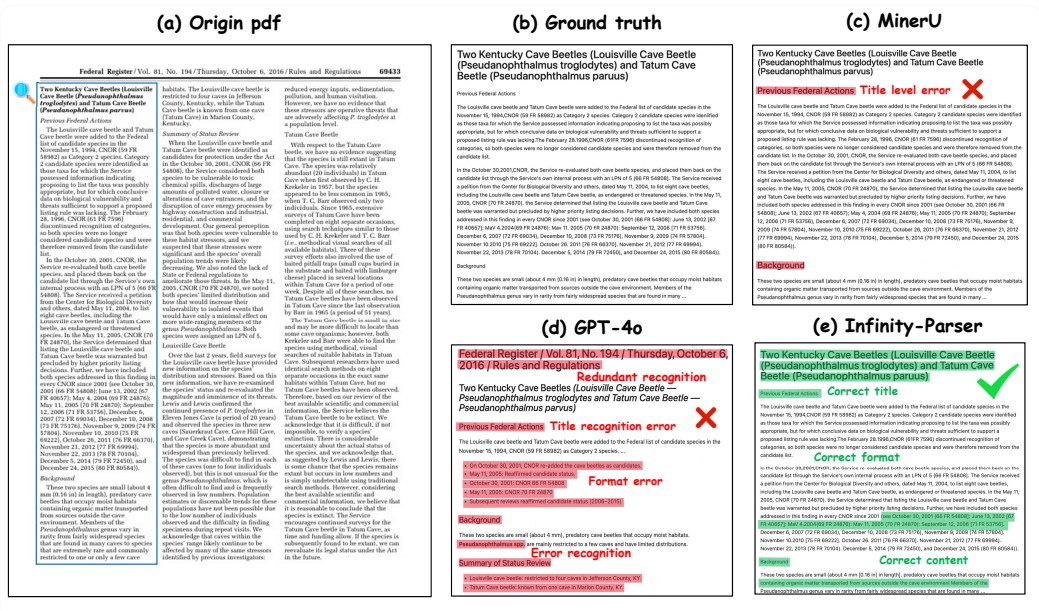

Figure 13: Comparison of Markdown extraction from a newspaper-style PDF. This figure presents the original densely formatted page (a), ground truth annotations (b), and the results from MinerU, GPT-4o, and Infinity-Parser (c–e). Due to the complex multi-column layout and title hierarchy, both MinerU and GPT-4o exhibit issues such as incorrect title levels, redundant content, and format errors. In contrast, Infinity-Parser accurately identifies the main title, maintains structural formatting, and preserves content integrity, demonstrating strong layout understanding in challenging document types.

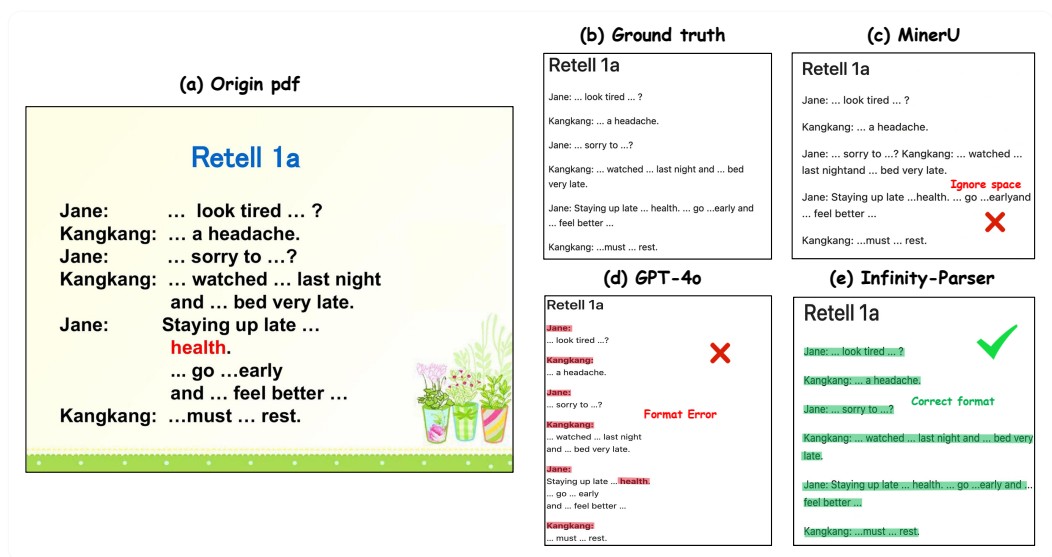

Figure 14: Comparison of Markdown extraction from a PowerPoint-style PDF slide. The figure includes the original slide (a), human-annotated ground truth (b), and outputs from MinerU, GPT-4o, and Infinity-Parser (c–e). MinerU and GPT-4o both struggle with layout fidelity, introducing spacing and formatting errors. In contrast, Infinity-Parser preserves the dialogue structure and formatting accurately, demonstrating its capability to handle informal, visually decorated slides effectively.