# OpenReview forum: "Infinity Parser: Layout Aware Reinforcement Learning for Scanned Document Parsing"
_ICLR.cc/2026/Conference — ICLR 2026 Conference Withdrawn Submission_

### Official Review · Reviewer_2Q4d · 2025-10-23

**Soundness:** 2
**Presentation:** 3
**Contribution:** 2
**Rating:** 4
**Confidence:** 3

**Summary:**

The paper introduces LayoutRL, a novel reinforcement learning framework for end-to-end parsing of scanned documents. To support reinforcement learning training, the authors also construct Infinity-Doc-400K, a large-scale dataset of ~400K scanned document pages paired with ground-truth Markdown parses. A vision-language model is trained is trained with LayoutRL on this data to yield Infinity-Parser, an end-to-end document parser.

**Strengths:**

1. The authors introduce Infinity-Doc-400K, a very large dataset. combining synthetic and pseudo-labeled real documents.
2. Infinity-Parser achieves state-of-the-art performance on multiple benchmarks.

**Weaknesses:**

1. The novelty of the paper is limited. GRPO optimization is already a part of any VLM architecture. The authors provide a task-specific loss, which we typically do during the retraining of a VLM on a particular task. It shouldn't be considered as the novelty of the work

2. In Table 2, the text edit and formula edit scores for English are lower than the Chinese ones. This suggests the training data of the Infinity parser is biased to the Chinese language.

3. The papers lack ablation. It only demonstrates how reinforcement learning is beneficial. It would be interesting to show the choice of VLM trained on Infinity-DOC-400K and the combination of loss functions.

4. As reported in Table 9 in the supplementary material, the method doesn't perform well on financial reports, academic papers, but performs well on notes and newspapers. However, the former ones have more structural representations than the latter ones. Then, how does the layout information help to generate structured machine translations?

5. In the introduction, it mentions the no. of documents is 400,482, then in the method, it says 400,066, and in the supplementary Figure 7, it says infinity doc 55K. I think the author should be consistent with the numbers.

**Questions:**

1. Why was only a 43K subset of the 400K dataset used for RL? Were the remaining documents reserved for testing/val, or was it a computational limitation? How might performance change if more data were used? Although the dataset comprises ~400,000 pages, the RL fine-tuning reportedly uses only a 43,000-page subset. It is unclear why the other data were not utilized and whether further gains could be achieved with full-scale RL training.

2. The three reward components are simply summed. Did you tune or validate these weights? Would different weights (or learned weights) affect the trade-off between local accuracy vs global structure?

**Details Of Ethics Concerns:**

The author already provided an ethical statement in the supplementary material.

---

> ### Author Response · Authors · 2025-11-24
> **Response to Reviewer 2Q4d (Part 1)**
>
> We sincerely appreciate your careful evaluation and valuable suggestions. Your feedback has been extremely helpful, and we respond to your points below.
>
> ---
>
>
> > **Q1: The novelty of the paper is limited. GRPO optimization is already a part of any VLM architecture. The authors provide a task-specific loss, which we typically do during the retraining of a VLM on a particular task. It shouldn't be considered as the novelty of the work.**
>
>
> Thank you for the reviewer’s thoughtful comments and for raising concerns about the novelty of our RL component. We understand the point that our method adopts the existing GRPO framework without introducing a fundamentally new optimization mechanism. However, our main contribution lies in showing that, even with almost no modification to the RL paradigm itself, carefully designing task-aligned, verifiable rewards based on edit distance and structural consistency, together with a lightweight adaptation of RL to a purely visual perception task such as document parsing, can lead to substantial and meaningful gains. Unlike reasoning-oriented RL methods that rely on explicit reasoning chains, document parsing involves no such latent reasoning process, making traditional “reasoning-enhanced RL” formulations inapplicable; this is precisely why our minimal but targeted adjustments are both necessary and effective. Despite their simplicity, these changes prove to be highly impactful: with only about 43K RL training samples, our approach surpasses contemporary SFT systems that rely on 4M–30M samples [1,2], demonstrating exceptional data efficiency and strong structural generalization. We believe that achieving such improvements without altering the underlying RL framework represents a valuable insight, offering a new perspective on applying RL to non-reasoning, structure-driven perception tasks. We appreciate the reviewer’s feedback, which has helped us clarify this contribution more explicitly in the revised version.
>
>
>
> > **Q2: In Table 2, the text edit and formula edit scores for English are lower than the Chinese ones. This suggests the training data of the Infinity parser is biased to the Chinese language.**
>
> Thank you for pointing this out. In the actual results, the English text-editing and formula-editing scores are *higher* than the corresponding Chinese scores, so the model does not exhibit weaker performance on English. It is true that our dataset contains slightly more Chinese documents than English, and we will include the exact language distribution in the revised version for completeness and transparency.
>
> > **Q3: As reported in Table 9 in the supplementary material, the method doesn't perform well on financial reports, academic papers, but performs well on notes and newspapers. However, the former ones have more structural representations than the latter ones. Then, how does the layout information help to generate structured machine translations?**
>
> Thank you for the reviewer’s question. We would like to clarify that the table in question evaluates an OCR-style task, which measures only the correctness of recognized textual content—i.e., whether characters are accurately extracted. This metric does not assess reading-order prediction, paragraph segmentation, or document-level structural hierarchy, which are precisely the structural abilities that layout information is intended to enhance. Therefore, improvements in reading-order coherence or paragraph boundary accuracy cannot be reflected in this table. The performance differences across document types are mainly due to the limitations of the base model itself. Importantly, despite these inherent constraints, our method still achieves substantial improvements over the base Qwen2.5-VL-7B model across all document categories, including significant error reductions on financial reports (0.194 → 0.070) and academic papers (0.195 → 0.087), as well as clear gains on notes (0.226 → 0.141) and newspapers (0.903 → 0.153).
>
>
> > **Q4: In the introduction, it mentions the no. of documents is 400,482, then in the method, it says 400,066, and in the supplementary Figure 7, it says infinity doc 55K. I think the author should be consistent with the numbers.**
>
> Thank you for pointing out the inconsistency in the reported document counts. This was indeed a typographical oversight. The correct number of documents is 400,482, and we will unify this value across the main paper and supplementary material in the revised version.

---

> ### Author Response · Authors · 2025-11-24
> **Response to Reviewer 2Q4d (Part 2)**
>
> > **Q5: It would be interesting to show the choice of VLM trained on Infinity-DOC-400K and the combination of loss functions.**
>
> Thank you for the reviewer’s insightful comment. Following the suggestion, we added an SFT-only setting trained on the full Infinity-Doc-400K dataset (Row 3 in the table). As expected, enlarging the SFT data from 43K to 400K leads to clear improvements across all metrics, confirming that simply scaling SFT does provide benefits. However, comparing Rows 6–8 reveals a more informative trend: even without any SFT, applying RL directly already yields substantial gains over the zero-shot baseline (e.g., Overall Cat. drops from 0.183 to 0.104, a relative reduction of ~43%). Building RL on top of the 43K SFT initialization further improves both structural and classification metrics. Most importantly, even when starting from the strongest 400K SFT-only baseline, adding RL still produces a significant improvement (Overall Cat. 0.096 vs. 0.122), indicating that a considerable portion of the performance gain must be attributed to the RL stage and its reward design rather than merely to scaling SFT data. We also observe that when the SFT dataset becomes very large, the model tends to overfit to the supervised signal, leaving less room for RL to further refine capabilities. This explains why the gains of 400K SFT+RL are slightly smaller than those of Zero-Shot+RL. While systematically studying the optimal data composition between SFT and RL in the post-training stage is valuable, it constitutes a separate line of investigation. In this paper, we focus on demonstrating that our reward-designed RL consistently and significantly improves generalization in document parsing tasks.
>
> | Method      | Edit Dist. | Count. | Order. | SFT  | RL   | Overall (EN)↓ | Overall (ZH)↓ | Overall Cat.↓ |
> |-------------|------------|--------|--------|------|------|----------------|----------------|----------------|
> | Zero Shot   | -          | -      | -      | -    | -    | 0.220          | 0.265          | 0.183          |
> | SFT         | -          | -      | -      | 43K  | -    | 0.198          | 0.261          | 0.159          |
> | SFT         | -          | -      | -      | 400K | -    | 0.154          | 0.240          | 0.122          |
> | Zero + RL   | ✔︎          | -      | -      | -    | 43K  | 0.169          | 0.224          | 0.156          |
> | Zero + RL   | ✔︎          | ✔︎      | -      | -    | 43K  | 0.159          | 0.200          | 0.112          |
> | Zero + RL   | ✔︎          | ✔︎      | ✔︎      | -    | 43K  | 0.141          | 0.197          | 0.104          |
> | SFT + RL    | ✔︎          | ✔︎      | ✔︎      | 43K  | 43K  | 0.163          | 0.195          | 0.092          |
> | SFT + RL    | ✔︎          | ✔︎      | ✔︎      | 400K | 43K  | 0.151          | 0.206          | 0.096          |
>
>
> > **Q6: Why was only a 43K subset of the 400K dataset used for RL? Were the remaining documents reserved for testing/val, or was it a computational limitation? How might performance change if more data were used? Although the dataset comprises ~400,000 pages, the RL fine-tuning reportedly uses only a 43,000-page subset. It is unclear why the other data were not utilized and whether further gains could be achieved with full-scale RL training.**
>
> Thank you for the question. The use of the 43K subset for the RL stage was initially constrained by computational limits, as RL training is significantly more expensive than SFT and we needed to evaluate multiple reward configurations; running full-scale RL on all 400K documents was not feasible. In the revised submission, we additionally include new experiments using the full 400K SFT-only model as well as the 400K SFT + 43K RL configuration. The expanded results show that scaling SFT from 43K to 400K improves the base model’s structural recovery ability, confirming the value of the full dataset itself

---

> ### Author Response · Authors · 2025-11-24
> **Response to Reviewer 2Q4d (Part 3)**
>
> > **Q7: The three reward components are simply summed. Did you tune or validate these weights? Would different weights (or learned weights) affect the trade-off between local accuracy vs global structure?**
>
>
> Thank you for raising this important point. We agree that different reward weightings could influence the balance between content fidelity and structural consistency, and we also consider a systematic exploration of a broader weighting space to be valuable. However, RL on long-document parsing is extremely resource-intensive, especially under a multi-rollout setting, and each new weighting configuration would require a full RL training run. This makes comprehensive weight tuning infeasible under our current computational budget. For this reason, we adopted the simplest and most stable setting—equal weights—which performed reliably in our preliminary experiments, and we focused our resources on analyzing the effectiveness of the reward components themselves. We plan to explore a larger reward-weighting search space in future work when computational resources allow, in order to gain a deeper understanding of how different rewards interact.
>
>
>
> ---
>
>
> Reference:
>
> [1] Z. Li et al., “MonkeyOCR: Document Parsing with a Structure-Recognition-Relation Triplet Paradigm,” [arXiv.org](http://arxiv.org/), 2025. https://arxiv.org/abs/2506.05218 (accessed Nov. 14, 2025).
>
> [2] H. Wei, Y. Sun, and Y. Li, “DeepSeek-OCR: Contexts Optical Compression,” [arXiv.org](http://arxiv.org/), 2025. https://arxiv.org/abs/2510.18234

---

### Official Review · Reviewer_DKbz · 2025-10-28

**Soundness:** 3
**Presentation:** 3
**Contribution:** 2
**Rating:** 4
**Confidence:** 4

**Summary:**

This paper introduces LayoutRL, a novel reinforcement learning framework designed to improve the parsing of scanned documents by making models explicitly layout aware. To achieve this, authors developed a multi-aspect reward model that evaluates content accuracy, paragraph segmentation, and reading order, moving beyond simple token-level supervision. Supporting this framework is the new, large-scale Infinity-Doc-400K dataset, which combines diverse synthetic and real-world documents. Infinity-Parser leverages this training to set a new state-of-the-art performance on various benchmarks, outperforming both specialized pipelines and general-purpose VLMs in accurately capturing complex document structures.

**Strengths:**

The paper introduces LayoutRL, an original and effective use of reinforcement learning with a multi-aspect reward model to explicitly teach models structural document layout, overcoming the generalization limits of standard fine-tuning.
It establishes a new state-of-the-art through exhaustive experiments across four diverse benchmarks, outperforming both specialized pipelines and general-purpose VLMs, thereby substantiating its claims with robust empirical evidence.
By creating and releasing the large-scale Infinity-Doc-400K dataset and the high-performance Infinity-Parser model, the work provides great resources that will accelerate future research and establish a new standard for the document AI community.

**Weaknesses:**

The paper does not investigate the relative importance of its different reward components (edit distances/paragraph counts/reading order) or adequately explain the negative interaction between SFT and RL, which does not provide deeper insights about why this framework works.

**Questions:**

If I’m correct, this paper only compares existing VLM baselines without fine-tuning them on the 400K Doc data. Including results after fine-tuning on different VLMs could better show the dataset’s significance.

---

> ### Author Response · Authors · 2025-11-24
> **Response to Reviewer DKbz (Part 1)**
>
> Thank you for your insightful and helpful review. We appreciate the time you dedicated to evaluating our work, and we address your concerns in the following response.
>
> ---
>
>
> > **Q1. The paper does not investigate the relative importance of its different reward components (edit distances/paragraph counts/reading order) or adequately explain the negative interaction between SFT and RL, which does not provide deeper insights about why this framework works.**
>
> Thank you for raising this point. Although the initial submission did not include reward-by-reward ablations, the results from both the original and the newly added experiments already provide meaningful insight into why the framework works. The three rewards target complementary aspects of structured document parsing: edit distance improves intra-paragraph fidelity, paragraph count prevents missing or hallucinated paragraphs, and reading order enforces global logical coherence. The substantial improvements observed after applying RL, including those in the newly added experiments, show that these components jointly contribute beyond what edit distance alone can achieve.  A systematic study of how SFT and RL should be combined in post-training—such as the optimal SFT scale, the training order, and the interaction between supervised signals and reward-driven updates—is indeed valuable but lies beyond the main focus of this paper, especially under space constraints. In this work, we focus on demonstrating that reward-designed RL reliably and significantly enhances structural generalization in document parsing, and we plan to explore the broader SFT–RL interplay more comprehensively in future work.
>
> > **Q2. This paper only compares existing VLM baselines without fine-tuning them on the 400K Doc data. Including results after fine-tuning on different VLMs could better show the dataset’s significance.**
>
> Thank you for the helpful suggestion. We acknowledge that we did not provide fine-tuning results on the full 400K-document dataset in the initial submission.We have now added experiments where both Qwen-2.5-VL 7B,  InternVL-3.5 8B and Llama-3.2-11B-Vision are fine-tuned on the 400K corpus. The updated results show clear and consistent improvements across all metrics for both models, indicating that the dataset substantially enhances systems with very different architectures and initial capabilities. These observations highlight the general usefulness of the 400K data rather than dependence on a specific backbone. We appreciate the reviewer’s suggestion, which helped strengthen this part of our evaluation.
>
>
>
> | Model               | Method    | Data Size | Overall (EN)↓ | Overall (ZH)↓ | Overall Cat.↓ |
> |---------------------|-----------|-----------|----------------|----------------|----------------|
> | Qwen-2.5-VL-7B-Instruct         | Zero Shot | -         | 0.220          | 0.265          | 0.183          |
> | Qwen-2.5-VL-7B-Instruct         | SFT       | 400K      | 0.154          | 0.240          | 0.122          |
> | InternVL-3.5-8B-Instruct        | Zero Shot | -         | 0.232          | 0.359          | 0.201          |
> | InternVL-3.5-8B-Instruct        | SFT       | 400K      | 0.171          | 0.286          | 0.162          |
> | Llama-3.2-11B-Vision-Instruct   | Zero Shot | -         | 0.448          | 0.760          | 0.640          |
> | Llama-3.2-11B-Vision-Instruct   | SFT       | 400K      | 0.216          | 0.392          | 0.246          |

---

### Official Review · Reviewer_b8ph · 2025-10-30

**Soundness:** 2
**Presentation:** 3
**Contribution:** 3
**Rating:** 4
**Confidence:** 4

**Summary:**

The paper proposes LayoutRL, an end-to-end reinforcement learning framework for layout-aware scanned document parsing. The authors build Infinity-Doc-400K, a large dataset combining synthetic pages and real-world pages. Trained on this data, Infinity-Parser achieves SOTA results across OmniDocBench, olmOCR-Bench, PubTabNet, and FinTabNet, with stronger OOD generalization and training stability than SFT baselines.

**Strengths:**

1. The paper presents LayoutRL, a complete reinforcement learning framework for document parsing with three well-designed reward functions that explicitly enhance the model’s understanding of document layout. The method is conceptually sound and experimentally comprehensive, covering multiple benchmarks and settings.

2. The authors introduce Infinity-Doc-400K, a large-scale 400K-document dataset built through multi-model collaborative annotation and template-based synthesis. This dataset provides a valuable and scalable foundation for future reinforcement learning research in layout-aware document understanding.

**Weaknesses:**

Weakness 1 – Limited analysis of experimental results
While the experiments are extensive, the analysis remains largely descriptive, focusing on trends and metrics rather than underlying causes. For instance, the claimed advantages of RL in stability and generalization are not sufficiently supported by detailed reasoning or ablation-based explanation.

Weakness 2 – Missing appendix
The paper repeatedly refers readers to an appendix for experimental settings and benchmark details, but no appendix is provided. This omission reduces the overall professionalism and completeness of the submission.

Weakness 3 – Insufficient experimental details
Key experimental configurations are missing, including training/testing splits, hyperparameters, and baseline evaluation dates. The lack of transparency limits reproducibility and makes it difficult to fully assess the reported results.

**Questions:**

1. The data construction section states that the authors built a 400K-document dataset, but the ablation studies repeatedly refer to experiments using 43K samples. Could the authors clarify how these two dataset scales are related and why only 43K samples were used for ablation?

2. Is the edit-distance reward computed over the entire document, or at a finer granularity (e.g., paragraph level)? If it is document-level, what is the specific motivation and necessity of the other two rewards (count and order)? If not, please provide a clearer description of the calculation scope.

3. In Table 5, the difference between English and Chinese edit distances varies dramatically between the second and third rows, while the average edit distance across document types remains almost unchanged; the opposite trend appears between the fourth and sixth rows. What factors contribute to these contrasting behaviors?

**Details Of Ethics Concerns:**

1. The paper states that real-world data were collected from sources such as financial reports, medical records, academic papers, books, magazines, and web pages. This raises potential copyright and data privacy concerns.

---

> ### Author Response · Authors · 2025-11-24
> **Response to Reviewer b8ph (Part 1)**
>
> Thank you for your thoughtful and constructive feedback. We appreciate your time and valuable comments. We provide clarifications to your concerns below.
>
> ---
>
> > **Weakness 1 – Limited analysis of experimental results While the experiments are extensive, the analysis remains largely descriptive, focusing on trends and metrics rather than underlying causes. For instance, the claimed advantages of RL in stability and generalization are not sufficiently supported by detailed reasoning or ablation-based explanation.**
>
> We thank the reviewer for highlighting the importance of deeper experimental analysis. To clarify the empirical evidence supporting the stability and generalization advantages of RL, we provide extensive structural-level visualizations in Appendix Figures 8–14. These examples illustrate typical failure patterns of SFT—such as page-level structural drift, hierarchical inconsistencies, and cross-region misalignment—and how RL effectively reduces these systematic errors by directly optimizing structural consistency. We would also like to clarify the roles of Figures 4 and 6. Figure 4 examines the *scaling behavior* of both SFT and RL: as training data increases, RL shows a steady and monotonic improvement, whereas SFT saturates early and exhibits instability at smaller scales, indicating its reliance on surface patterns rather than underlying structural rules. Figure 6 further reinforces this observation: under the in-distribution setting, SFT maintains stable paragraph-level accuracy but plateaus early on page-level accuracy, while RL continues to improve. Under the out-of-distribution setting, the contrast becomes even sharper—SFT degrades substantially on unseen document types, whereas RL maintains strong performance, demonstrating superior structural generalization. These pieces of evidence together provide a detailed and multi-level explanation for the advantages attributed to RL.
>
> > **Weakness 2 – Missing appendix The paper repeatedly refers readers to an appendix for experimental settings and benchmark details, but no appendix is provided. This omission reduces the overall professionalism and completeness of the submission.**
>
> This may stem from a misunderstanding about the formatting requirements of ICLR: the appendix must be submitted as a *separate Supplementary Material file* rather than appended directly to the main manuscript. We have already uploaded the complete supplementary document—including experimental settings, benchmark descriptions, data details, and ablations—to the system’s Supplementary Material section, and the main paper includes explicit references to it. We also note that other reviewers have asked questions based on the appendix content, suggesting that the supplementary file is fully accessible. We appreciate the reviewer’s attention to the completeness of the submission.
>
> > **Weakness 3 – Insufficient experimental details Key experimental configurations are missing, including training/testing splits, hyperparameters, and baseline evaluation dates. The lack of transparency limits reproducibility and makes it difficult to fully assess the reported results.**
>
> We appreciate the reviewer’s attention to reproducibility. All experimental configurations are provided in the Supplementary Material, including the training/validation splits, input-length distributions, dataset statistics (Appendix D), the full set of document types (Appendix E), descriptions of all benchmarks (Appendix F), and complete training and implementation details (Appendix G), covering batch sizes, optimization settings, rollout configurations, evaluation procedures, and the exact versions of all baseline models. Since these details are organized within the supplementary file, they may not have been immediately visible in the main manuscript. We will improve navigation and referencing in the camera-ready version to make these components easier to locate.
>
>
>
>
>
> > **Q4: The data construction section states that the authors built a 400K-document dataset, but the ablation studies repeatedly refer to experiments using 43K samples. Could the authors clarify how these two dataset scales are related and why only 43K samples were used for ablation?**
>
> Thank you for the question. Although our full dataset contains roughly 400K documents, running repeated SFT/RL training and ablations on the entire corpus would be computationally prohibitive. To ensure fair and comparable experiments, we therefore random sampled a 43K subset covering the major document types and structural patterns, and used it consistently across all SFT, RL, and ablation settings. As shown in Figure 1, RL scales smoothly with data size, and its performance at 43K already matches or exceeds public SOTA systems, indicating that this subset is sufficient to validate the method and reflect the benefits of our data design. We will make the distinction between the full dataset and the 43K subset clearer in the revised version.

---

> ### Author Response · Authors · 2025-11-24
> **Response to Reviewer b8ph (Part 2)**
>
> > **Q5: Is the edit-distance reward computed over the entire document, or at a finer granularity (e.g., paragraph level)? If it is document-level, what is the specific motivation and necessity of the other two rewards (count and order)? If not, please provide a clearer description of the calculation scope.**
>
> We appreciate the reviewer’s attention to the granularity of our reward design. The edit-distance reward is *not* computed over the entire document directly. Instead, we first perform paragraph-level matching between the model outputs and the target document using Hungarian matching to identify one-to-one paragraph correspondences. Before matching, both the model output and the ground truth are split into paragraphs using the standard \n\n delimiter. Edit distance is then computed within each matched paragraph pair and averaged. This ensures that the edit-distance reward focuses on intra-paragraph textual fidelity without conflating it with paragraph ordering.The paragraph-count reward constrains the model to avoid hallucinating or missing paragraphs, and the reading-order reward explicitly enforces global logical ordering. Together, these three rewards cover the key dimensions of structured document parsing: internal paragraph correctness, structural completeness, and global reading flow. Additional illustrative examples are provided in the appendix.
>
> > **Q6:  In Table 5, the difference between English and Chinese edit distances varies dramatically between the second and third rows, while the average edit distance across document types remains almost unchanged; the opposite trend appears between the fourth and sixth rows. What factors contribute to these contrasting behaviors?**
>
> Thank you for the perceptive observation. The contrasting trends arise because the three metrics correspond to two fundamentally different tasks. Overall (EN)*{Edit} and Overall (ZH)*{Edit} are structural metrics for document parsing, and are highly sensitive to reading order, paragraph boundaries, and multi-column layout. In contrast, Overall_{Cat} is an OCR-level metric that only evaluates whether local text tokens are correctly recognized, independent of structural restoration. For example, in a two-column PDF, the first line of column A and the first line of column B may appear visually adjacent, causing OCR to treat them as sequential text; however, they are not adjacent in logical reading order, making this distinction crucial for document parsing. When only the edit-distance reward is added, RL substantially improves structural recovery, leading to large reductions in EN/ZH edit distances, while the OCR-oriented Overall_{Cat} remains largely unchanged. Conversely, from Row 4 to Row 6, adding SFT primarily strengthens token-level imitation within sentences, which benefits OCR-style metrics more directly; thus Overall_{Cat} improves more significantly, while structurally driven edit-distance metrics improve less. This difference reflects the varying sensitivities of the two task types and directly motivates our reward design.

---

> ### Author Response · Authors · 2025-11-24
> **Response to Reviewer b8ph (Part 3)**
>
> > **Q7:  The paper states that real-world data were collected from sources such as financial reports, medical records, academic papers, books, magazines, and web pages. This raises potential copyright and data privacy concerns.**
>
> We thank the reviewer for highlighting these important issues. We would like to clarify that our ethical considerations and data-handling practices are explicitly described in **Appendix B (Ethics Statement)**. As stated there, our work does not involve identifiable human subjects, personal user data, or sensitive user information. The Infinity-Doc-400K dataset combines automatically generated synthetic documents (via HTML-based rendering) with real-world samples that are pseudo-labeled through cross-model agreement and manually filtered for quality. For domains such as financial reports and web pages, we only use content that is legally accessible for non-commercial research, and any future release will be strictly under an academic license and limited to annotations, aggregate statistics, synthetic data, or clearly license-compliant subsets, rather than raw copyrighted content. The medical records used in our dataset come from real clinical cases but are provided to us **only after strict de-identification** by the data provider, with all direct and quasi-identifiers removed; we do not have access to hospital databases or identifiable health information in the sense of GDPR, and we do not plan to release any raw clinical documents or page images. We also respect website terms of service and robots.txt when collecting web data and apply filters to remove obvious PII when detected. We believe, and have argued in Appendix B, that under these safeguards our work poses minimal risk of discrimination, bias, or privacy violations, while aiming to improve the robustness and reliability of document parsing technologies. In the camera-ready version, we will make these points more prominent in the main text to avoid any ambiguity.

---

### Official Review · Reviewer_uxVc · 2025-11-01

**Soundness:** 2
**Presentation:** 3
**Contribution:** 2
**Rating:** 4
**Confidence:** 3

**Summary:**

The paper proposes LayoutRL, a layout-aware RL framework for end-to-end scanned document parsing. It trains a VLM to output the final structured document and scores it with three automatic rewards — edit distance, paragraph-count consistency, and reading-order preservation — so the model learns structure, not just tokens. A new 400K document dataset (Infinity-Doc-400K) is built to make this RL feasible.

**Strengths:**

1.Motivation is clear and well grounded (SFT struggles on page-level / OOD structure).

2.Writing is clear and figures are intuitive, so the method is easy to follow.

**Weaknesses:**

1.The methodological novelty is somewhat concentrated on constructing a large, structurally aligned corpus and on formulating task-specific, verifiable rewards; the RL component itself follows existing group-relative / rule-based RLFT paradigms and does not introduce a fundamentally new optimization mechanism.

2.A key experiment is missing, namely training Qwen2.5-VL-7B on the proposed Infinity-Doc-400K dataset without applying the RL stage, in order to explicitly verify how much of the performance gain should be attributed to reinforcement learning.

**Questions:**

None

---

> ### Author Response · Authors · 2025-11-24
> **Response to Reviewer uxVc**
>
> Thank you for your thoughtful and constructive comments. We sincerely appreciate your feedback and address your points below.
>
>
> ---
>
> > **1. The methodological novelty is somewhat concentrated on constructing a large, structurally aligned corpus and on formulating task-specific, verifiable rewards; the RL component itself follows existing group-relative / rule-based RLFT paradigms and does not introduce a fundamentally new optimization mechanism.**
>
>
>
> Thank you for the reviewer’s thoughtful comments and for raising concerns about the novelty of our RL component. We understand the point that our method adopts the existing GRPO framework without introducing a fundamentally new optimization mechanism. However, our main contribution lies in showing that, even with almost no modification to the RL paradigm itself, carefully designing task-aligned, verifiable rewards based on edit distance and structural consistency, together with a lightweight adaptation of RL to a purely visual perception task such as document parsing, can lead to substantial and meaningful gains. Unlike reasoning-oriented RL methods that rely on explicit reasoning chains, document parsing involves no such latent reasoning process, making traditional “reasoning-enhanced RL” formulations inapplicable; this is precisely why our minimal but targeted adjustments are both necessary and effective. Despite their simplicity, these changes prove to be highly impactful: with only about 43K RL training samples, our approach surpasses contemporary SFT systems that rely on 4M–30M samples [1,2], demonstrating exceptional data efficiency and strong structural generalization. We believe that achieving such improvements without altering the underlying RL framework represents a valuable insight, offering a new perspective on applying RL to non-reasoning, structure-driven perception tasks. We appreciate the reviewer’s feedback, which has helped us clarify this contribution more explicitly in the revised version.
>
>
>
>
> ---
>
>
> > **2. A key experiment is missing, namely training Qwen2.5-VL-7B on the proposed Infinity-Doc-400K dataset without applying the RL stage, in order to explicitly verify how much of the performance gain should be attributed to reinforcement learning.**
>
>
> Thank you for the reviewer’s insightful comment. Following the suggestion, we added an SFT-only setting trained on the full Infinity-Doc-400K dataset (Row 3 in the table). As expected, enlarging the SFT data from 43K to 400K leads to clear improvements across all metrics, confirming that simply scaling SFT does provide benefits. However, comparing Rows 6–8 reveals a more informative trend: even without any SFT, applying RL directly already yields substantial gains over the zero-shot baseline (e.g., Overall Cat. drops from 0.183 to 0.104, a relative reduction of ~43%). Building RL on top of the 43K SFT initialization further improves both structural and classification metrics. Most importantly, even when starting from the strongest 400K SFT-only baseline, adding RL still produces a significant improvement (Overall Cat. 0.096 vs. 0.122), indicating that a considerable portion of the performance gain must be attributed to the RL stage and its reward design rather than merely to scaling SFT data. We also observe that when the SFT dataset becomes very large, the model tends to overfit to the supervised signal, leaving less room for RL to further refine capabilities. This explains why the gains of 400K SFT+RL are slightly smaller than those of Zero-Shot+RL. While systematically studying the optimal data composition between SFT and RL in the post-training stage is valuable, it constitutes a separate line of investigation. In this paper, we focus on demonstrating that our reward-designed RL consistently and significantly improves generalization in document parsing tasks.
>
>
> ---
>
>
> | Method      | Edit Dist. | Count. | Order. | SFT  | RL   | Overall (EN)↓ | Overall (ZH)↓ | Overall Cat.↓ |
> |-------------|------------|--------|--------|------|------|----------------|----------------|----------------|
> | Zero Shot   | -          | -      | -      | -    | -    | 0.220          | 0.265          | 0.183          |
> | SFT         | -          | -      | -      | 43K  | -    | 0.198          | 0.261          | 0.159          |
> | SFT         | -          | -      | -      | 400K | -    | 0.154          | 0.240          | 0.122          |
> | Zero + RL   | ✔︎          | -      | -      | -    | 43K  | 0.169          | 0.224          | 0.156          |
> | Zero + RL   | ✔︎          | ✔︎      | -      | -    | 43K  | 0.159          | 0.200          | 0.112          |
> | Zero + RL   | ✔︎          | ✔︎      | ✔︎      | -    | 43K  | 0.141          | 0.197          | 0.104          |
> | SFT + RL    | ✔︎          | ✔︎      | ✔︎      | 43K  | 43K  | 0.163          | 0.195          | 0.092          |
> | SFT + RL    | ✔︎          | ✔︎      | ✔︎      | 400K | 43K  | 0.151          | 0.206          | 0.096          |

---

> > ### Author Response · Authors · 2025-11-24
> > **Response to Reviewer uxVc (Part 2)**
> >
> > Reference:
> >
> > [1] Z. Li et al., “MonkeyOCR: Document Parsing with a Structure-Recognition-Relation Triplet Paradigm,” [arXiv.org](http://arxiv.org/), 2025. https://arxiv.org/abs/2506.05218 (accessed Nov. 14, 2025).
> >
> > [2] H. Wei, Y. Sun, and Y. Li, “DeepSeek-OCR: Contexts Optical Compression,” [arXiv.org](http://arxiv.org/), 2025. https://arxiv.org/abs/2510.18234

---

### Note · Authors · 2025-12-25

**Comment:**

I would like to withdraw my submitted manuscript.
Thank you for your understanding.

**Withdrawal Confirmation:**

I have read and agree with the venue's withdrawal policy on behalf of myself and my co-authors.